# Closing the Modality Gap Aligns Group-Wise Semantics

**Eleonora Grassucci**[*] **Giordano Cicchetti**[*]**, Emanuele Frasca, Aurelio Uncini,**
**Danilo Comminiello**
Department of Information Engineering, Electronics, and Telecommunications
Sapienza University of Rome
`{name}.{surname}@uniroma1.it`    [*]Authors contributed equally

## Abstract

In multimodal learning, CLIP has been recognized as the *de facto* method for learning a shared latent space across multiple modalities, placing similar representations close to each other and moving them away from dissimilar ones. Although CLIP-based losses effectively align modalities at the semantic level, the resulting latent spaces often remain only partially shared, revealing a structural mismatch known as the modality gap. While the necessity of addressing this phenomenon remains debated, particularly given its limited impact on instance-wise tasks (e.g., retrieval), we prove that its influence is instead strongly pronounced in group-level tasks (e.g., clustering). To support this claim, we introduce a novel method designed to consistently reduce this discrepancy in two-modal settings, with a straightforward extension to the general $n$-modal case. Through our extensive evaluation, we demonstrate our novel insight: while reducing the gap provides only marginal or inconsistent improvements in traditional instance-wise tasks, it significantly enhances group-wise tasks. These findings may reshape our understanding of the modality gap, highlighting its key role in improving performance on tasks requiring semantic grouping. Code available at: `https://github.com/ispamm/ModGap`.

## 1 Introduction

The canonical goal of contrastive representation learning is to learn information from the original data by embedding semantically similar data points nearby and dissimilar ones far apart. Multimodal research has inherited this goal, assuming that a video, its caption, and its soundtrack should share the same neighborhood in a joint latent space. Unfortunately, the multimodal latent space shows different behavior, and representations tend instead to preserve their modality cluster, hindering the semantic alignment. This phenomenon, known as the modality gap Liang et al. (2022), is prevalent in all multimodal models grounded on the conventional and widespread contrastive InfoNCE loss function van den Oord et al. (2018) like CLIP Radford et al. (2021). Before training, samples from the same modality initially cluster together due to different random model weights initialization, forming distinct modality-specific groups. Unfortunately, these clusters persist even after training, resulting in a sparse and fragmented latent space.

In recent works, the modality gap has been studied and partially addressed for image and text pairs Eslami & de Melo (2025); Yaras et al. (2025); Fahim et al. (2024); Schrodi et al. (2025); Mistretta et al. (2025). Concurrently, other works are instead

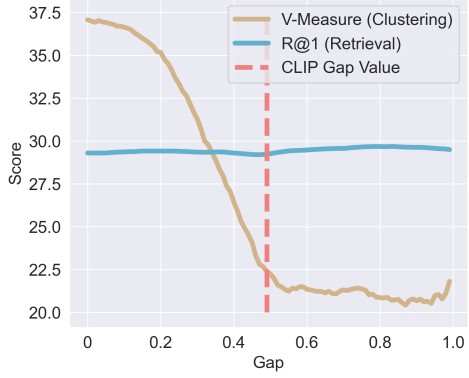

Figure 1: Reducing the gap consistently improves clustering metrics, while leaving unaffected retrieval ones. On the contrary, increasing the gap downgrades the V-Measure, bringing no improvements in R@1. In CLIP, the gap results in very poor clustering performance due to the latent space fragmentation.

accepting the gap, letting the space as it is while focusing on hyperbolic similarities Ramasinghe et al. (2024). However, the reasons for mitigating or untouching such a gap are not well-grounded and the advantages in common downstream tasks such as retrieval are inconsistent throughout the literature. Yaras et al. (2025) argues that closing the gap may improve the retrieval performance in downstream tasks, while Schrodi et al. (2025) found that a larger modality gap has a mild positive correlation with downstream performance. Moreover, all these approaches draw conclusions focusing only on the two-modal case (mainly image and text), without advancing cues in the case of multiple modalities.

Rather than following conventional views, we reconsider the modality gap in the context of group-wise tasks. We provide evidence that the modality gap is irrelevant for instance-wise tasks, such as retrieval, which depend on relative rankings, but strongly impacts multimodal group-wise tasks, such as clustering, which rely on absolute distances among representations of multimodal data in the latent space. Specifically, we show that closing the modality gap reduces the within-group scatter, leading to more coherent semantic groupings and better clustering performance, while leaving retrieval rankings mostly unaffected, as it is clear from Fig. 1. To address this, we propose a novel and effective objective function that explicitly encourages the alignment of matching pairs while maintaining a uniform and expressive latent structure. We show that our method effectively reduces the modality gap in both bimodal and trimodal benchmarks, improves clustering performance by several points in V-Measure, and maintains or slightly increases retrieval accuracy. These results confirm that the modality gap is not a harmless artifact, but a central factor shaping the geometry of the multimodal latent space. Crucially, we show that it can be effectively minimized through a simple objective, enabling better semantic organization without compromising instance-level precision.

Our main contributions can be summarized as follows.

- We demonstrate that the modality gap, while irrelevant to instance-wise tasks such as retrieval, directly impacts group-wise semantics reflecting in tasks, like clustering, that require explicitly closing the gap.
- We introduce a novel objective that combines true-pair alignment and centroid-based uniformity to effectively close the modality gap. Our method scales to multiple modalities and requires no architectural changes or post-hoc corrections.
- Across both bimodal and trimodal benchmarks, our approach reduces the gap while improving clustering and preserving retrieval performance, confirming its effectiveness for multimodal representation learning.

## 2 RELATED WORK

**Multimodal Learning.** Starting from CLIP Radford et al. (2021) several multimodal models have been developed for two modalities like CLAP Elizalde et al. (2023) or CLIP4Clip Luo et al. (2021). Lately, the same InfoNCE loss van den Oord et al. (2018) has been extended to multiple modalities in ImageBind Girdhar et al. (2023) or VAST Chen et al. (2023b). More recently, novel approaches have been proposed for multimodal learning to avoid the cosine similarity loss, namely GRAM Cicchetti et al. (2025) and Symile Saporta et al. (2024).

**Modality Gap.** The modality gap has been observed for the first time by Liang et al. (2022), and then studied mainly for the CLIP model Wu et al. (2023); Fahim et al. (2024); Shi et al. (2023) or for generic image and text pairs Mistretta et al. (2025); Yaras et al. (2025); Schrodi et al. (2025); Wu et al. (2023); Zhang et al. (2023). These works provide theoretical justification for the gap and propose to mitigate the gap by fixing the temperature Yaras et al. (2025), by applying post-hoc translations in the latent space Liang et al. (2022); Schrodi et al. (2025), or by sharing the transformer encoder and the projection layer in the vision and language encoders Eslami & de Melo (2025). In any case, each of these methods studied the modality gap in the case of two modalities, without advancing clues on the case of three or more modalities.

**Effect of Temperature in Contrastive Learning.** The temperature $\tau$ in the InfoNCE loss van den Oord et al. (2018) is recognized among the primary knobs for steering what features a contrastive learner captures. SimCLR Chen et al. (2020) noted that smaller $\tau$ sharpens the softmax and boosts instance-level retrieval accuracy. Wang & Isola (2020) proved that $\tau$ trades off alignment against uniformity, while higher $\tau$ instead favors cluster or class structure. Gradient-based analyses later showed that $\tau$ effectively controls the penalty on hard negatives, with low values concentrating

gradients on the most confusable samples Wang & Liu (2020). Building on this view, Kukleva et al. (2023) introduced a schedule between high-$\tau$ (group-wise) and low-$\tau$ (instance-wise) phases. Most recently, Dinu et al. (2025) showed that increasing $\tau$ compresses embeddings, lowering intrinsic dimensionality while preserving task performance, suggesting a second-order role for $\tau$ in model size and deployment efficiency.

## 3 WHY CLOSING THE GAP?

### 3.1 PRELIMINARIES

We consider a training batch $\mathcal{B} = \{(\mathbf{x}_i^1, \mathbf{x}_i^2, \ldots, \mathbf{x}_i^M)\}_{i=1}^N$, where each sample is observed across $M$ modalities, indexed by $m = \{1, 2, \ldots, M\}$. For each modality $m$, the encoder $f^m : \mathcal{X}^m \to \mathbb{R}^d$ maps the input $\mathbf{x}_i^m$ into a $d$-dimensional embedding vector $\mathbf{z}_i^m = f^m(\mathbf{x}_i^m)$, that is further normalized to obtain a unitary norm vector. We use $\text{sim}(\mathbf{z}_i^m, \mathbf{z}_j^n)$ to denote the cosine similarity between embeddings from modality $m$ and $n$, i.e., $\text{sim}(\mathbf{z}_i^m, \mathbf{z}_j^n) = \langle \mathbf{z}_i^m, \mathbf{z}_j^n \rangle$.

Given a pair of modalities $(m, n)$ with $m \neq n$, we define the InfoNCE loss for modality pair as:

$$\mathcal{L}^{(m \to n)} = -\frac{1}{N} \sum_{i=1}^N \log \frac{\exp\left(\text{sim}(\mathbf{z}_i^m, \mathbf{z}_i^n)/\tau\right)}{\sum\limits_{j=1}^N \exp\left(\text{sim}(\mathbf{z}_i^m, \mathbf{z}_j^n)/\tau\right)}, \tag{1}$$

where $\tau > 0$ is the temperature parameter controlling the softness of the distribution over negatives. The total contrastive loss is defined by averaging the two directions of the InfoNCE loss, $\mathcal{L}^{(m \to n)}$ and $\mathcal{L}^{(n \to m)}$. This is the standard loss used in CLIP Radford et al. (2021) and its variants for different modalities. The temperature $\tau$ modulates the contribution of hard negatives Kukleva et al. (2023). A low temperature penalizes harder negatives (i.e., those with high similarity to the anchor), favoring instance-level discrimination. In contrast, a higher temperature distributes gradients more evenly across all negatives, which typically promotes the emergence of semantic clusters and benefits group-wise reasoning tasks Dinu et al. (2025). In multimodal learning, different approaches have been proposed to handle the temperature effect including learning it during training or fixing at predefined values. Throughout the paper, we refer to methods with the standard InfoNCE loss with learnable temperature as CLIP (LT) and with fixed temperature as CLIP (FT).

### 3.2 UNDERSTANDING THE MODALITY GAP

A gap among modalities exists at the initialization phase, where different encoders initialized with random weights represent data in different narrow cones in the shared latent space Liang et al. (2022). Nevertheless, the gap persists even during the entire contrastive training phase. What is more, such contrastive learning dynamics have been recognized to be the root cause of the gap, regardless of the initialization. Therefore, even though the final learned space is somewhat semantically aligned, positive pairs are decoupled and very distant, as Fig. 6 in the Appendix shows. As demonstrated by Shi et al. (2023); Cicchetti et al. (2025), the traditional CLIP loss function easily gets stuck in local minima, in which positive pairs are somewhat matched but far from each other, fostering the modality gap.

In detail, previous works show that the conventional CLIP loss function is composed of two terms, each with specific objectives Wang & Isola (2020); Shi et al. (2023): a first term tries to align positive pairs, while the second one tries to spread away non-matching pairs. In practice, these two terms provide opposite contributions, resulting in balanced and opposite forces. Therefore, models easily end up in local minima, avoiding the gap closure while allowing the representations of the two modalities to align with each other in "semantic stripes". These semantic stripes allow however good performance in retrieval tasks since positive pairs have higher cosine similarity with respect to non-matching pairs, even though such similarity is far from the ideal 1.0, as we show in Section 5. Therefore, representations of matching pairs do not lie in the same portion of the latent space and are instead quite far from each other, severely limiting the expressiveness and the group-wise alignment of the latent space.

Following Liang et al. (2022), to measure such a gap between two generic modalities $m$ and $n$, we measure the effective Euclidean distance between the centroids of each modality:

$$\text{Gap}_{m,n} = \|\mathbf{c}^m - \mathbf{c}^n\|, \tag{2}$$

where $\mathbf{c}^m = \frac{1}{N} \sum_{i=1}^{N} \mathbf{z}_i^m$. Even though the gap is zero, this does not imply that the embeddings are effectively aligned in the latent space. Therefore, we further adopt the mean cosine similarity true pairs metric, defined as:

$$\text{Cos TP}_{m,n} = \frac{1}{N} \sum_{i=1}^{N} \langle \mathbf{z}_i^m, \mathbf{z}_i^n \rangle. \tag{3}$$

This metric measures how much the normalized matching pairs are near each other in the latent space. Intuitively, the closer to 1.0, the smaller the angle is between them, and the closer the matching pairs lie in the latent space, being more aligned.

### 3.3 ALIGNING GROUP-WISE SEMANTICS

While the modality gap does not necessarily disrupt instance-wise semantics, which depend on relative similarity rankings, it may prevent effective structuring of the latent space for group-wise objectives such as clustering. Let us formalize the reason starting from the InfoNCE loss in equation 1. Such a loss directly optimizes the relative ordering between the similarity of the true pair $\text{sim}(\mathbf{z}_i^m, \mathbf{z}_i^n)$ and those of negatives $\text{sim}(\mathbf{z}_i^m, \mathbf{z}_j^n)$, $j \neq i$. The condition for retrieval success (e.g., Recall@K) is satisfied whenever:

$$\text{sim}(\mathbf{z}_i^m, \mathbf{z}_i^n) > \max\left(\text{sim}(\mathbf{z}_i^m, \mathbf{z}_j^n)\right), \quad \forall j \neq i. \tag{4}$$

Thus, as long as InfoNCE ensures this inequality, the relative ranking is preserved, regardless of whether absolute similarity values are close to $1.0$ or far from it. Since instance-wise tasks like retrieval depend only on relative ordering and not on the absolute placement of embeddings, they are insensitive to the modality gap.

For group-wise tasks, let us build the centroid of semantic class $c$ comprising two modalities $m$ and $n$ in the following way:

$$\boldsymbol{\mu}_s^\delta = \frac{1}{2}\left((\mathbf{z}_s^m + \boldsymbol{\delta}) + (\mathbf{z}_s^n + \boldsymbol{\delta})\right), \tag{5}$$

where the modalities are shifted according to a constant vector $\boldsymbol{\delta}$, representing the gap impact between the two modalities. Note that, in the case of 0 gap, the formula trivially resolves in $\boldsymbol{\mu}_s^0 = \frac{1}{2}(\mathbf{z}_s^m + \mathbf{z}_s^n)$. The expected within-class scatter for a generic modality $(m)$ decomposes as

$$\mathbb{E}_s\left[\|\mathbf{z}_s^m - \boldsymbol{\mu}_s^\delta\|_2^2\right] \approx \mathbb{E}_s\left[\|\mathbf{z}_s^m - \boldsymbol{\mu}_s^0\|_2^2\right] + \|\boldsymbol{\delta}\|^2, \tag{6}$$

with the gap term $\boldsymbol{\delta}$ summed to the expectation since it does not depend on the semantics of class $s$, since it is orthogonal to the span of semantic vectors and therefore constant for each of them, as proven by Zhang et al. (2023). Therefore, enlarging the gap uniformly inflates every semantic cluster, degrading homogeneity and completeness, while shrinking the gap tightens such clusters.

**Intuitive consequence.** The modality gap leaves rank-based, instance-wise decisions mostly untouched, as they are the primary objective of the InfoNCE loss, but widens absolute distances and thereby harms any objective that optimizes within/between-cluster geometry. Retrieval is therefore almost flat across the gap value, whereas clustering (and other group-wise tasks) benefit from driving the gap to zero, as shown in Fig. 1. For instance, suppose to retrieve a cat image caption, the sufficient condition is that the caption remains the single most similar text, irrespective of its absolute cosine value. Instead, if the goal is forming a "cat" cluster, the necessary condition is that every cat image–caption pair lies close to the "cat" centroid, and a gap between the two modalities inflates that radius, so cluster purity drops even though each individual pair is still top-ranked for retrieval.

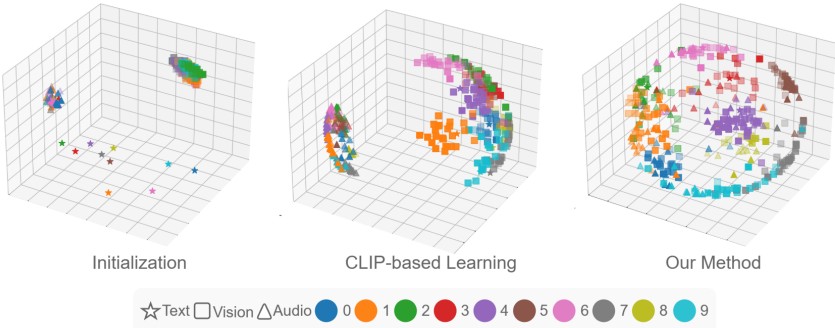

Figure 2: AV-MNIST multimodal latent space. The CLIP-based learning creates a fragmented latent space with embeddings clearly clustered by modality and not by multimodal semantics. Our method closes the gap and enhances group-wise semantics, placing embeddings of the same class in the same portion of the space, effectively learning a semantically meaningful multimodal latent space.

## 4    CLOSING THE MODALITY GAP

We aim to close the modality gap while ensuring consistent alignment among the positive pair distribution. To achieve such scope, we propose two novel losses. The first one, the Align True Pairs loss $\mathcal{L}_{\text{ATP}}$ guarantees the alignment between true pairs. Considering $M$ modalities, among which $\mathbf{a}$ is the anchor one (i.e., the modality to which all the other modalities are aligned to Girdhar et al. (2023); Zhu et al. (2024)), the loss is:

$$\mathcal{L}_{\text{ATP}} = \frac{1}{M-1} \sum_{m \in M, m \neq a} \left( \frac{1}{N} \sum_{i=1}^{N} \left( ||\mathbf{z}_i^m - \mathbf{z}_i^a||_2^2 \right) \right), \tag{7}$$

where $m$ is taken from the set of available modalities, $\mathbf{a}$ is the anchor modality and $N$ is the batch size. The second one, the Centroid Uniformity loss $\mathcal{L}_{\text{CU}}$, ensures uniformity of the semantic classes among the modalities in the latent space by:

$$\mathcal{L}_{\text{CU}} = \log \left( \frac{1}{N} \sum_{i=1}^{N} \sum_{j=1, j \neq i}^{N} \exp \left( -2||\boldsymbol{\mu}_i - \boldsymbol{\mu}_j||_2^2 \right) \right), \tag{8}$$

in which $\boldsymbol{\mu}_k$, with $k = i, j$, are the centroids defined as:

$$\boldsymbol{\mu}_k = \frac{1}{M} \sum_{m \in M} \mathbf{z}_k^m, \tag{9}$$

and $\mathbf{c}_k$ is the centroid of the $k$-th element of the batch built by averaging all the modalities embeddings. The effect of the two losses is complementary. The $\mathcal{L}_{\text{ATP}}$ promotes closeness between positive pairs, effectively enhancing the mean cosine similarity between them. However, involving solely such a loss may produce a side effect: the entire latent space collapses into small portions, placing representations of dissimilar data in the same portion of the latent space, as we show in the Appendix. Therefore, the contribution of the $\mathcal{L}_{\text{CU}}$ loss becomes crucial, ensuring the sparsification of the latent space by enforcing uniformity to the centroids while preserving the learned alignment. Indeed, moving the centroids of matching pairs in the space implies moving the modalities representations accordingly, effectively preserving alignment while leveraging the whole space. Without centroids, uniformity should have been applied independently to each modality as in Wang & Isola (2020), disrupting the learned alignment among similar semantic pairs, as we show in the Appendix. Additionally, the radial basis function (RBF) kernel in equation 8 is well related to the uniform distribution on the unit hypersphere where multimodal representations lie Wang & Isola (2020), therefore enforcing the coverage of the whole surface of the hypersphere. The overall proposed loss function that aims at aligning the true pairs and closing the modality gap is a sum of the two terms:

$$\mathcal{L}_{\text{gap}} = \mathcal{L}_{\text{ATP}} + \mathcal{L}_{\text{CU}}. \tag{10}$$

Such loss should be then combined with the contrastive loss to obtain:

$$\mathcal{L}_{\mathrm{CL_{gap}}} = \mathcal{L}_{\mathrm{gap}} + \frac{1}{2}\left(\mathcal{L}^{(m\rightarrow n)} + \mathcal{L}^{(n\rightarrow m)}\right), \tag{11}$$

for the two-modality $m$ and $n$ case to lighten the notation, but it can be expanded to more modalities.

## 5 EXPERIMENTAL VALIDATION

We perform extensive evaluations in controlled and real-world scenarios with four different datasets, four tasks, and a diverse number of modalities.

### 5.1 SETTING

To evaluate the proposed method in multimodal scenarios, we design a series of experiments that progressively increase in complexity and scale. Following the literature, we begin with two foundational experiments using the CIFAR-10 and AV-MNIST datasets, which involve two and three modalities, respectively. Subsequently, we scale up our experiments using the MSCOCO and MSR-VTT datasets, which offer more complex multimodal data.

**CIFAR10** (Image-Text). The CIFAR10 dataset consists of 60k $32\times32$ color images evenly distributed across 10 classes, with 50k images for training and 10k for testing Krizhevsky et al. (2009). For our experiments, we pair each image with its corresponding class label as text, creating a bimodal dataset. For CIFAR10, we use two separate ResNet50 encoders, one for text and the other for images.

**AV-MNIST** (Audio-Visual-Text). AV-MNIST is a synthetic dataset combining visual, auditory, and textual modalities. It is the union of two well-known datasets: the MNIST dataset composed of 60k $28\times28$ digit images and the Audio-MNIST Becker et al. (2023) containing 30k audio samples of spoken digits (0-9) from diverse speakers. Each sample is also associated with a textual label that represents the digit. Encoder details in the Appendix.

**MSCOCO** (Image-Caption). The MSCOCO dataset contains over 330k images, each annotated with five human-generated captions, totaling more than 1.5M captions Lin et al. (2014), and it is a well-known benchmark for analyzing the modality gap. To scale up the model capabilities and test our approach in this challenging dataset, we use EVA-CLIP ViT-G Sun et al. (2023) as the visual encoder and BERT-B Devlin et al. (2019) to process the textual captions.

**MSR-VTT** (Video-Audio-Text). The MSR-VTT dataset comprises 10k video clips spanning 20 categories, with each clip annotated with approximately 20 natural language sentences, resulting in a total of 200k captions Xu et al. (2016). Each video clip includes visual frames, audio tracks, and corresponding textual descriptions, making it a trimodal dataset. Each sample in the dataset is associated to one of the 20 categories. This makes the dataset useful to be evaluated with clustering metrics. We maintain the same architectures used for MSCOCO plus BEATs Chen et al. (2023a), a transformer-based model, used to extract features from the audio tracks.

### 5.2 TASKS

We conduct a suite of experiments covering both instance-wise and group-wise tasks.

**Cross-Modal (CM) Retrieval.** We evaluate the model ability to associate related data across modalities (e.g., text to audio or text to image) with cross-modal retrieval. For a given query in one modality, the task is to retrieve matching items from another modality using cosine similarity in the shared latent space. This task quantifies the instance-level alignment and cross-modal discrimination capacity of the learned embeddings.

**Clustering.** We assess the structural organization of the latent space using supervised and unsupervised clustering. We apply standard clustering algorithms (k-means and k-NN) and evaluate the resulting clusters using V-Measure and k-NN accuracy. High performance in clustering indicates that the latent space preserves semantic coherence across modalities and aligns samples belonging to the same class or coarse-grained semantic group closely, regardless of the input domain.

**Multimodal (MM) Retrieval with Cross-Conditioning.** We evaluate the performance of downstream models that use the extracted embeddings from the modality encoders following Li et al.

(2021). For this task, we involve an additional multimodal encoder that takes as input all the modalities and fuses them. Therefore, a proper alignment is crucial in this task. Such a multimodal encoder outputs the probability score of the semantic match among them. To this purpose, following Chen et al. (2023b), as the multimodal downstream encoder we reuse the text encoder attaching an MLP on top of it to extract the probability value. Along with our brand-new loss functions, we add the data-anchor matching (dam) loss function to train these components, as in Li et al. (2021); Chen et al. (2023b); Cicchetti et al. (2025):

$$\mathcal{L}_{\mathrm{dam}} = \mathbb{E}_{(\mathbf{a},\mathbf{m}_2,\cdots,\mathbf{m}_k)\sim(A,M_2,\cdots,M_k)}\left[y\log p_{\mathrm{dam}} + (1-y)\log(1-p_{\mathrm{dam}})\right]. \tag{12}$$

This task stresses the ability of the encoders to build coherent, modality-agnostic representations and provides insight into the expressiveness and compositionality of the learned space. Higher performance in this task highlights an increased facility by the multimodal encoder to detect interdependencies among modalities.

**Captioning.** A downstream task to evaluate the benefit of an aligned and well-structured latent space is data captioning. Following Yan et al. (2022), we involve a decoder serving as the language generator model and we add a specific loss term that, along with our proposed loss functions, trains the encoder-decoder structure for this specific task. Intuitively, the more aligned and semantically coherent the latent space, the better the model will generate the captions. The captioning loss is:

$$\mathcal{L}_{\mathrm{cap}} = -\sum_{t=1}^{T} \log P_\theta(\mathbf{y}_t|\mathbf{y}_{<t},\mathbf{x}), \tag{13}$$

where $\mathbf{y}$ is the exact tokenized text the model aims to learn by maximizing the conditional likelihood under the forward autoregressive factorization. $P_\theta(\mathbf{y}_t|\mathbf{y}_{<t},\mathbf{x})$ denotes the probability assigned to the token $\mathbf{y}_t$ given as input the past history $\mathbf{y}_{<t}$ and the input features $\mathbf{x}$. Captioning requires both accurate semantic grounding and compositional generalization, and therefore serves as a complementary proxy for evaluating the usefulness of the learned features in downstream generative tasks.

## 5.3 RESULTS

**Aligning group-wise semantics**

Figure 1 shows that increasing the modality gap leads to a drop in clustering metrics, despite having a negligible effect on retrieval scores, while decreasing it considerably improves the clustering metric (+17.5 points). This suggests that instance-level alignment is tolerant to cross-modal offsets, while group-wise metrics benefit from a more globally structured latent space. Crucially, **the optimal clustering performance coincides with the zero-gap** configuration, indicating that full semantic overlap facilitates more coherent clusters. The plot is obtained by starting from the gap value of standard CLIP on the MSR-VTT dataset and then posthoc shifting

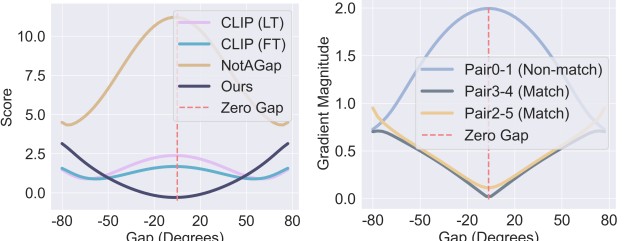

Figure 3: **(a)** Following Liang et al. (2022), we place six simulated image-text embedding pairs on a 3D sphere, with two mismatched pairs. We artificially move these pairs toward closing or enlarging the gap among them and we track the loss landscape during the simulation. **(b)** During the same simulation we keep track also of the gradient magnitude received by the six embeddings pairs through our design loss function $\mathcal{L}_{\mathrm{CL_{gap}}}$. When the gap is closer to zero, the contribution to the loss is just matter of the non matching pairs.

the matching pairs up to 0.0 and 1.0 gap. Second, the simulation in Figure 3(a) shows that, in presence of data mismatching, our **proposed loss induces a landscape in which the global minimum is achieved when the modality gap is zero**. In contrast, standard contrastive losses such as CLIP achieve the minimum around 60 degrees, thus when matching pairs are still quite far in the latent space. Proof in the Appendix. Therefore, we can train the model with the proposed losses to reach the zero modality gap while aligning group-wise semantics and improving the clustering performance without affecting retrieval performance. The underlying mechanism is clarified in Figure 3(b), which shows that **as the modality gap narrows, the gradient magnitude associated with non-matching**

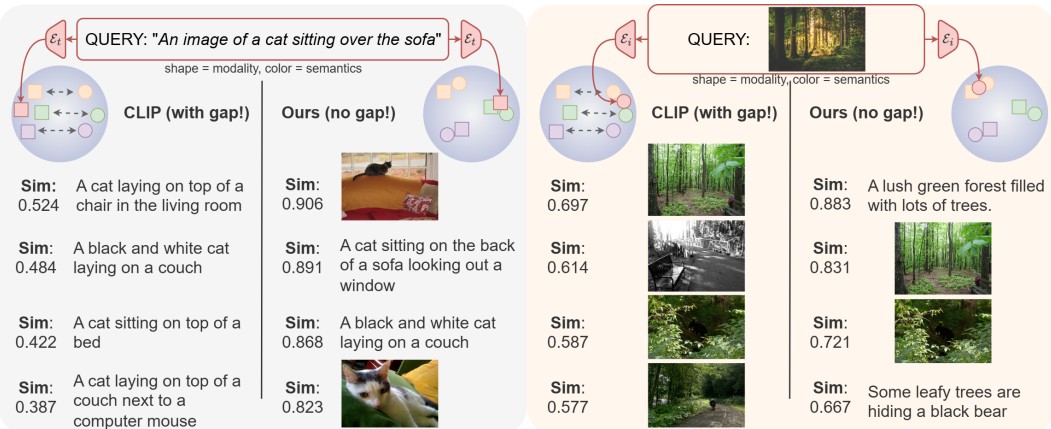

Figure 4: Most similar vectors from the MSCOCO vector databases with both modalities embedded. CLIP learned space has a gap of 0.47 and, for a text query, all the closest embeddings come from the text modality, meaning that embeddings are clustered according to the modality and not according to the overall multimodal semantics. On the contrary, the proposed method has nearly a zero gap, and most similar samples come from both the image and text modalities, proving that we can effectively build a multimodal latent space that is semantically coherent among the modalities. The same behavior holds for an image query.

**pairs increases**, while that of matching pairs decreases. With a large gap, non-matching samples provide the same gradient contribution as matching ones. As the gap closes, these negatives become harder and more semantically informative, thus dominating the optimization process and encouraging the model to better shape the space by adjusting the missing matches. This naturally improves the structuring of semantic clusters without altering the rank of positive pairs (which have nearly zero gradients with the closed gap), hence preserving (or improving) retrieval performance. Altogether, these observations highlight that minimizing the modality gap leads to more informative gradient signals by emphasizing semantically meaningful negatives, thereby improving the alignment and group-wise structure of the latent space. This reflects in improved clustering performance without affecting retrieval. By explicitly encouraging the zero-gap configuration, our proposed loss $\mathcal{L}_{\text{CL}_{\text{gap}}}$ provides a straightforward method to close the gap and align group-wise semantics.

Figure 2 shows the latent space plot in the case of embedding dimension equal to 3 in the AV-MNIST dataset. Notably, the plotted spaces are not stochastically generated but the real latent space in $\mathbb{R}^3$. As it is clear, the proposed method closes the gap better than conventional CLIP-based learning. Crucially, this consistently improves group-wise clustering, placing embeddings from the same class closer to each other in a semantically meaningful latent space.

Table 1 quantitatively proves our claims on the relation between the gap and the group-wise semantics. Indeed, as the gap is progressively closed with the proposed method, the clustering performance crucially improves, while retrieval accuracy between text and video (TV) and text and audio (TA) is barely affected by the gap. Overall, this intuition is congruous across diverse datasets with different numbers (two and three) and types of modalities (image, video, text, audio), even though the simpler the dataset, the easier is for the model to align group-wise semantics.

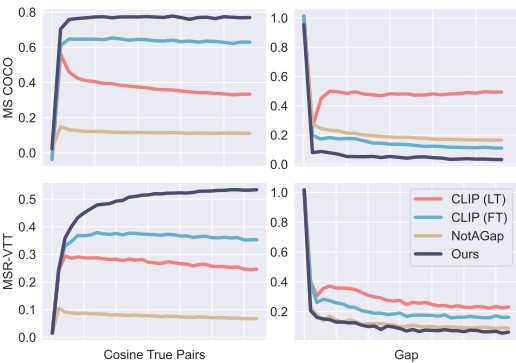

Figure 5: CosTP and gap during training.

**Closing the gap**

Figure 4 shows the most similar samples by querying the learned latent spaces of CLIP and of our method without forcing the query to be

Table 1: Instance-wise (cross-modal (CM) retrieval) vs group-wise (clustering) tasks correlated to the gap value. The harder the dataset, the more evident the results.

| | | | CM Retrieval | | Clustering | |
|---|---|---|---|---|---|---|
| Method | Dataset | Gap ↓ | TV R@1 | TA R@1 | V-Measure | kNN |
| CLIP (LT) (Radford et al., 2021) | CIFAR10 (2 modal) | 0.86 | 82.0 | - | 67.0 | 81.2 |
| CLIP (FT) (Yaras et al., 2025) | | 0.14 | 82.1 | - | 67.6 | 81.9 |
| Ours | | **0.09** | **82.4** | - | **67.9** | **82.4** |
| CLIP (LT) Radford et al. (2021) | MSCOCO (2 modal) | 0.47 | **74.6** | - | 12.98 | 26.3 |
| CLIP (FT) Yaras et al. (2025) | | 0.12 | 73.2 | - | 12.99 | 31.0 |
| Ours | | **0.03** | 70.3 | - | **23.63** | **36.4** |
| CLIP (LT) (Radford et al., 2021) | AV-MNIST (3 modal) | 0.20 | 87.1 | 84.2 | 77.6 | 87.0 |
| CLIP (FT) (Yaras et al., 2025) | | 0.24 | 84.1 | 80.4 | 73.8 | 85.0 |
| Ours | | **0.09** | **88.7** | **89.1** | **82.7** | **89.2** |
| CLIP (LT) (Radford et al., 2021) | MSR-VTT (3 modal) | 0.29 | 34.2 | 10.3 | 23.3 | 52.9 |
| CLIP (FT) (Yaras et al., 2025) | | 0.19 | **34.8** | 10.1 | 31.3 | 55.7 |
| Ours | | **0.07** | 32.8 | **11.8** | **32.1** | **58.0** |

Table 2: Multimodal (MM) retrieval and captioning results.

| | | Space Measures | | MM Retrieval | Captioning | | |
|---|---|---|---|---|---|---|---|
| Method | Dataset | Gap ↓ | Cos TP ↑ | R@1 | BLEU@1 | BLEU@3 | CIDEr |
| CLIP (LT) | MSCOCO (2 modal) | 0.47 | 0.34 | 72.5 | 45.8 | 22.5 | 153.2 |
| CLIP (FT) | | 0.12 | 0.63 | 73.8 | 45.9 | 23.0 | 155.0 |
| NotAGap | | 0.17 | 0.11 | 75.6 | 45.4 | 22.2 | 153.3 |
| Ours | | **0.03** | **0.77** | **77.3** | **46.1** | **23.2** | **160.9** |
| CLIP (LT) | MSR-VTT (3 modal) | 0.24 | 0.27 | 30.6 | 26.7 | 9.5 | 63.6 |
| CLIP (FT) | | 0.17 | 0.37 | 30.3 | 26.2 | 9.4 | 63.2 |
| NotAGap | | 0.09 | 0.07 | 29.9 | 24.3 | 8.7 | 53.8 |
| Ours | | **0.06** | **0.53** | **33.3** | **26.8** | **9.6** | **64.4** |

cross-modal by construction. This means that the similarity cares about the semantics only, regardless of the modality. Interestingly, when querying the CLIP space for the nearest vectors, while returning semantically similar samples, the embeddings always belong to the same modality as the query. In contrast, in the gap-free latent space learned with our method, the closest vectors not only share the same semantics but also come from different modalities. This proves that closing the modality gap effectively helps build a semantically aligned multimodal latent space.

Table 2 compares the proposed method and the current literature in closing the gap and downstream tasks. As it is clear from Tab. 2, our novel loss closes the multimodal gap by achieving nearly 0.0 distance between the modality clusters centroids. Additionally, we better align true pairs by crucially increasing their cosine similarity from 0.34 of the standard CLIP (LT) to 0.77 in MS COCO. The capabilities of modeling the multimodal latent space are evident in Fig. 5 too, where the proposed method outperforms the conventional CLIP, other solutions with fixed temperature (FT) Yaras et al. (2025), and NotAGap Fahim et al. (2024). Importantly and counter-intuitively, during training, the cosine similarity of true pairs slightly but progressively decreases in comparison configurations. This means that the true pairs are progressively worse aligned. On the contrary, the proposed method considerably increases the cosine similarity between true pairs, truly aligning them, while also definitely closing the gap among modalities, regardless of whether there are two or three modalities.

## 6 CONCLUSION

In this work, we revisited the modality gap in multimodal representation learning, providing insights into the impact of this phenomenon on downstream tasks. Through theoretical analysis and extensive experiments across bimodal and trimodal datasets, we showed that while the modality gap has limited influence on instance-level tasks such as retrieval, it significantly affects group-wise tasks like clustering. To address this, we proposed a novel objective that explicitly aligns true pairs while promoting latent space uniformity. Our method consistently reduces the modality gap and improves clustering performance without compromising retrieval accuracy, offering a simple yet effective solution for better semantic organization in multimodal spaces.

## ACKNOWLEDGEMENTS

This work was partially supported by the European Union under the NRRP of NextGeneration-EU, partnership on "Future Artificial Intelligence Research" (PE00000013 – SPOKE 5 - CUP B53C22003980006 - FAIR: High Quality AI) and partnership on "Telecommunications of the Future" (PE00000001 - program "RESTART"), and partially by the *Progetti di Ateneo* of Sapienza University of Rome under grant RM123188F75F8072 and RM1241910FC4BEEA, and partially by the Italian Ministry of University and Research (MUR) within the PRIN 2022 Program for the project "EX-EGETE: Explainable Generative Deep Learning Methods for Medical Signal and Image Processing", under grant number 2022ENK9LS, CUP B53D23013030006.

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

APPENDIX

A.1 PROOFS AND MOTIVATIONS

**The gap is nearly the same for all the pairs.** We empirically prove our theoretical claims and we compute both the Euclidean distance and the cosine similarities among true matching pairs, which could both be interpreted as gap measures. As it is clear from Tab. 3, the variance in the gap is extremely low and nearly 0, so all the points have the same gap. Thus, it is reasonable and empirically-grounded to assume the same $\boldsymbol{\delta}$ as a measure for the gap in Sec. 4, or to use the gap measured as centroid distance in equation 2, being it a single measure for the overall gap. The same findings can be applied to equation 6 as well. Concurrently, Zhang et al. (2023) demonstrated that the gap vector is orthogonal to the text and image embeddings, and that the gap can be approximated with a constant vector, confirming our findings.

Table 3: Mean and variance of the distance between the pairs in MSCOCO and MSR-VTT datasets.

| Dataset | Distance Mean | Distance Var | Cos TP Mean | Cos TP Var |
|---|---|---|---|---|
| MSCOCO (T-I) | 1.153 | 0.004 | 0.334 | 0.005 |
| MSR-VTT (T-V) | 1.163 | 0.009 | 0.319 | 0.012 |
| MSR-VTT (T-A) | 1.252 | 0.010 | 0.211 | 0.016 |

**Proof for global minimum in Fig. 3.** The 60 minimum in Fig. 3 is a direct outcome of the controlled synthetic setup and the dynamics of the InfoNCE loss $\mathcal{L}^{(m \rightarrow n)}$. In the experiment, image ($\mathbf{v}$) and text ($\mathbf{t}$) embeddings lie on two fixed rings with an initial angular offset of $\Delta = 120$. During the optimization, the true image–text pairs are brought closer (current gap = $\theta$), while the swapped pairs (i.e., the negatives) remain at their original positions. Therefore, from the image anchor perspective, these negatives are now at an angular distance of $\Delta - \theta$. This geometric property is exact and does not rely on any symmetry or regular placement.

The InfoNCE loss balances attraction to the true pair and repulsion from negatives. Its derivative becomes zero when the similarity to the true caption matches the similarity to the closest negative (this occurs when $\theta = \Delta/2$). In our test case, that means CLIP converges to 60, as observed in Fig. 3. Importantly, this also implies that InfoNCE never drives $\theta$ to 0, regardless of the initial gap $\Delta$.

In contrast, our method includes the explicit $\mathcal{L}_{\text{ATP}}$ term that always decreases as $\theta \rightarrow 0$, and does not depend on the position of the negatives. Formally, our loss can be written as $\mathcal{L}_{\text{CL}_{\text{gap}}} = \mathcal{L}^{(m \rightarrow n)} + \mathcal{L}_{\text{ATP}} + \mathcal{L}_{\text{CU}}$. The $\mathcal{L}_{\text{ATP}}$ term is $\mathcal{L}_{\text{ATP}} \propto ||\mathbf{v} - \mathbf{t}||_2^2 = 2(1 - \cos\theta)$, whose gradient is $\partial\mathcal{L}_{\text{ATP}} = 2\sin\theta$, always pulling the pair toward perfect overlap ($\theta = 0$) and has no counteracting repulsion. The $\mathcal{L}_{\text{CU}}$ term depends only on the class centroids, not on the intra-pair angle, so $\partial\mathcal{L}_{\text{CU}} = 0$. Therefore, the total derivative becomes

$$\frac{\partial\mathcal{L}_{\text{CL}_{\text{gap}}}}{\partial\theta} = \frac{\partial\mathcal{L}^{(m \rightarrow n)}}{\partial\theta} + 2\sin\theta, \tag{14}$$

with the extra term $2\sin\theta$ shifting the sole stationary point to $\theta = 0$, which is now a global minimum. This holds for any initial configuration.

**Proof for equation 6.** For a semantic class $s$ and two modalities $m, n$, the class centroid under a uniform shift $\delta$ is equation 5):

$$\boldsymbol{\mu}_s^\delta = \frac{1}{2}\big((\mathbf{z}_s^m + \boldsymbol{\delta}) + (\mathbf{z}_s^n + \boldsymbol{\delta})\big) = \boldsymbol{\mu}_s^0 + \boldsymbol{\delta}, \tag{15}$$

where

$$\boldsymbol{\mu}_s^0 = \frac{1}{2}\left(\mathbf{z}_s^m + \mathbf{z}_s^n\right). \tag{16}$$

We expand the squared deviation:

$$\begin{aligned}
\left\| \mathbf{z}_s^m - \boldsymbol{\mu}_s^{\boldsymbol{\delta}} \right\|_2^2 &= \left\| \mathbf{z}_s^m - (\boldsymbol{\mu}_s^0 + \boldsymbol{\delta}) \right\|_2^2 \\
&= \left\| (\mathbf{z}_s^m - \boldsymbol{\mu}_s^0) - \boldsymbol{\delta} \right\|_2^2 \\
&= \left\| \mathbf{z}_s^m - \boldsymbol{\mu}_s^0 \right\|_2^2 + \left\| \boldsymbol{\delta} \right\|_2^2 - 2 \left\langle \mathbf{z}_s^m - \boldsymbol{\mu}_s^0, \ \boldsymbol{\delta} \right\rangle.
\end{aligned} \tag{17}$$

Taking the expectation over classes yields:

$$\mathbb{E}_s \left[ \left\| \mathbf{z}_s^m - \boldsymbol{\mu}_s^{\boldsymbol{\delta}} \right\|_2^2 \right] = \mathbb{E}_s \left[ \left\| \mathbf{z}_s^m - \boldsymbol{\mu}_s^0 \right\|_2^2 \right] + \left\| \boldsymbol{\delta} \right\|_2^2 - 2 \, \mathbb{E}_s \left[ \left\langle \mathbf{z}_s^m - \boldsymbol{\mu}_s^0, \ \boldsymbol{\delta} \right\rangle \right]. \tag{18}$$

Since $\boldsymbol{\delta}$ is approximately constant across classes and orthogonal to the span of semantic vectors Zhang et al. (2023), the cross-term vanishes:

$$\mathbb{E}_s \left[ \left\langle \mathbf{z}_s^m - \boldsymbol{\mu}_s^0, \ \boldsymbol{\delta} \right\rangle \right] \approx 0. \tag{19}$$

Thus, we obtain equation 6:

$$\mathbb{E}_s \left[ \left\| \mathbf{z}_s^m - \boldsymbol{\mu}_s^{\boldsymbol{\delta}} \right\|_2^2 \right] \approx \mathbb{E}_s \left[ \left\| \mathbf{z}_s^m - \boldsymbol{\mu}_s^0 \right\|_2^2 \right] + \left\| \boldsymbol{\delta} \right\|_2^2. \tag{20}$$

## A.2 EXPERIMENTS DETAILS

To process the AudioMNIST dataset, we compute the Mel spectrograms with 128 nmels, fmax at 8000, hop length equal to 512, and 2048 nfft. As image encoder, we employ a convolutional neural network (CNN) comprising a two-layer convolutional architecture with 32 and 64 filters, respectively, ReLU activation functions, Max Pooling, and a final MLP layer to map the features into the latent space. For the audio modality, we design a three-layer convolutional encoder with 16, 32, and 64 filters, ReLU activations, and an MLP layer to similarly project audio features into the latent space.

**Clustering on MSCOCO.** MSCOCO contains the information about objects that are present inside an image, and each image could contain more than one object, up to an undefined upper bound (also 10 different objects in an image), making standard clustering hard to apply. However, we devise a pipeline to perform cross-modal clustering on this dataset. We take into consideration images from the test dataset that contain only a single object, so that we have a univocal correspondence between the image and a single textual caption representing such an object. We report clustering results on MSCOCO in Tab.1.

**Computing resources.** The AV-MNIST experiments are conducted on an RTX4080 with 16GB. The CIFAR10 experiments on an A6000 with 48GB. The experiments on MS COCO and MSR-VTT experiments on a node with $4 \times$ A100.

## A.3 ADDITIONAL EXPERIMENTS

We visualize the PCA latent spaces for two (MSCOCO) and three modalities (MSR-VTT) at initialization, after the conventional CLIP-based training, and after the training with the proposed method. Figure 6 shows the results. From Fig. 6, the gap closure is evident. Contrary to CLIP-based learning, our method can close the gap and spread modality embeddings leveraging the whole hypersphere space. Interestingly, Fig. 6 shows a similar behavior of the gap for the two- and three-modal case. At initialization, one modality tends to occupy more space, being more sparse, while the other/others are much more grouped. With CLIP-based learning, the modalities still tend to similarly lie in the space. However, regardless of the initialization and of the number of modalities, the proposed method completely closes the gap, building a more compact and well-aligned space.

**Additional gap measures.** Together with the gap formulation in equation 2, we further measure the gap as suggested in Schrodi et al. (2025), where the authors proposed the Relative Modality Gap (RMG) measure. We report the results in Tab. 4.

**Closing the gap zero-shot.** We consider the models trained on MS COCO and then evaluate them in a zero-shot fashion on the validation set of OpenImage-V7. The entire validation set contains 41620 images, each of them has associated one class among more than 20k image classes. We evaluate the

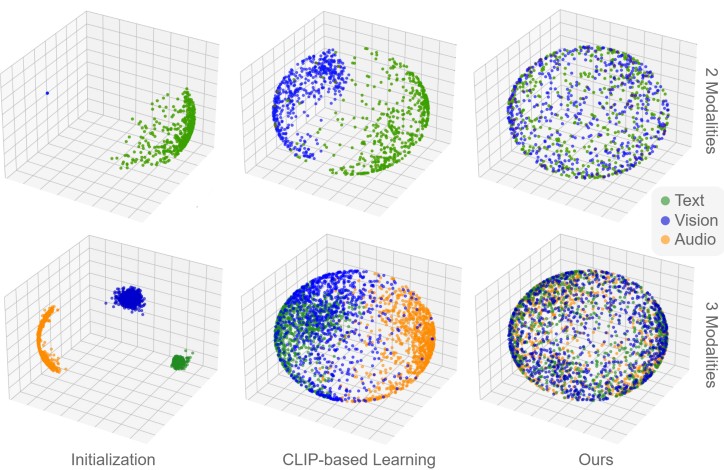

Figure 6: Left, latent spaces at initialization with the modality clusters clearly defined. Center, learned after training from CLIP-based learning that preserves the modality gaps. Right, space learned with our method, covering the whole space with overlapping clusters. PCA visualizations.

Table 4: Relative Modality Gap (RMG) measure for the methods on MSCOCO and MSR-VTT datasets.

|            | MSCOCO | MSR-VTT (T-V) | MSR-VTT (T-A) |
|------------|--------|---------------|---------------|
| CLIP (LT)  | 0.43   | 0.44          | 0.48          |
| CLIP (FT)  | 0.27   | 0.40          | 0.46          |
| Ours       | **0.17** | **0.33**    | **0.44**      |

ability of the proposed loss combinations to zero-shot closing the gap on the whole validation set, measuring the gap as in equation 2, the Cosine True Pairs (Cos TP) as in equation 3, and also the Relative Modality Gap (RMG) suggested in Schrodi et al. (2025). The results in Tab. 5 shows that the proposed method well-scale to large-scale datasets, achieving consistent results in all the metrics and outperforming the comparisons by considerably reducing the modality gap (absolute and relative). Furthermore, we also evaluate the clustering performance of the methods on this dataset. For this purpose, we select from the validation set the images containing one category among the 10 more common classes: 'Plant', 'Car', 'Person', 'Clothing', 'Food', 'Flower', 'Tree', 'Mammal', 'Wheel', 'Dog'. Moreover, we rebalance the dataset so that, in the final version, each class contains an equal number of samples. In this experiment too, the V-Measure achieved by the proposed method is the highest with respect to other methods, proving the effectiveness of the proposed method once again.

Table 5: Zero-shot gap and clustering results on OpenImage-V7 validation set.

| Method    | V-Measure ↑ | Gap ↓  | RMG ↓  | Cos TP ↑ |
|-----------|-------------|--------|--------|----------|
| CLIP (LT) | 16.7        | 0.459  | 0.462  | 0.210    |
| CLIP (FT) | 13.5        | 0.374  | 0.516  | 0.224    |
| Ours      | **17.2**    | **0.311** | **0.429** | **0.273** |

**Relation between gap and clustering performance.** To further highlight the relation between clustering performance and the gap, we plot the gap measure and the V-Measure during the training of the ResNet50 encoders on the CIFAR10 dataset. Fig 7 shows the results. As the gap decreases, the V-Measure evaluations increase reaching the best score with the proposed method, which better closes the gap.

**Impact of the initialization.** According to Liang et al. (2022) that first discovered the modality gap phenomenon, the gap exists at initialization and it is then preserved by the conventional contrastive loss. Nevertheless, the key reason for the modality gap is the contrastive behavior of the InfoNCE

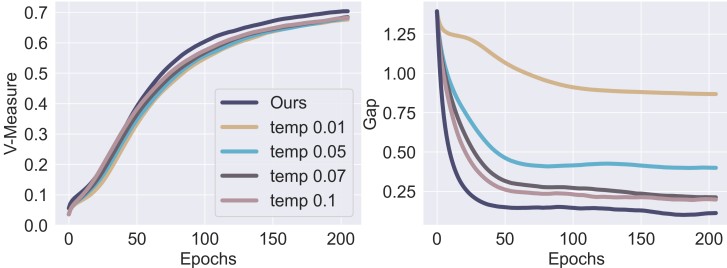

Figure 7: Clustering performance (V-Measure) and gap measure during training on the CIFAR10 dataset.

loss and not the initialization. To prove it, we compare two configurations with the same setup, model (ResNet50), and loss (InfoNCE). We enforce an initial sparsification of the learned space for the first epoch so that the encoders first learn to sparsify the embeddings, regardless of the modality. We do so by using as loss the uniformity loss over the hypersphere by Wang & Isola (2020). However, once the encoders learn the sparsification and we activate instead the conventional contrastive learning loss, the encoders start to separate the modalities again and the gap begins to be recreated, as we show in Fig. 8. We conduct the same experiment with the proposed losses to close the gap, and no effect is revealed with the initial sparsification as well. Therefore, although the gap is created at initialization, this does not impact the learning procedure and it is therefore the conventional contrastive loss function that tends to create such a gap.

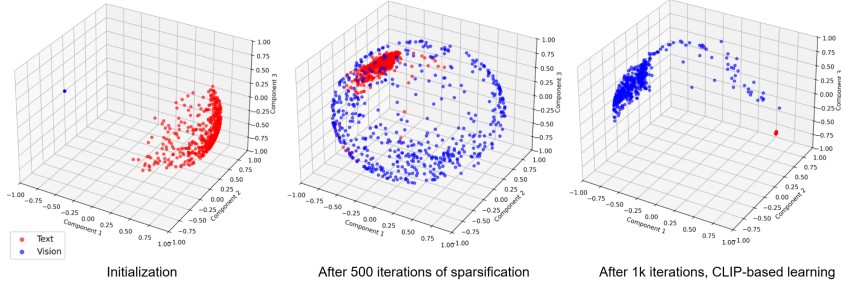

Figure 8: Latent space visualization during training of the conventional InfoNCE loss. The gap is created at initialization, as already known in Liang et al. (2022). However, even if we initially train the models to learn to sparsify the latent space (second plot) and reduce the gap, as soon as we reintroduce the conventional InfoNCE loss function, the gap is recreated (third plot). Experiment with ResNet50 from scratch on MS COCO dataset.

**Experiments on the MultiBench classification task.** We perform additional experiments with comparison methods in a different task. We rerun the experiments using CoMM Dufumier et al. (2025) repository in three datasets (MOSI, UR-FUNNY, MuSTARD) for the classification task from MultiBench. Then, we apply on CoMM the method proposed in Yaras et al. (2025) for a further comparison (CoMM + Yaras et al. (2025)). Finally, we add our proposed losses to the CoMM framework (CoMM + Ours line in the table below). We report gap and accuracy measures. As it is clear from Tab. 6, the proposed loss combination improves results in terms of accuracy with respect to CoMM and reduces the gap in all the datasets.

**Visualizing the semantic stripes.** In the presence of the modality gap, embeddings tend to form "semantic stripes". Such stripes allow good performance in retrieval tasks since positive pairs have higher cosine similarity with respect to non-matching pairs, even though such similarity is far from the ideal 1.0, but low performance in group-wise tasks as embeddings coming from the same semantic concept lie separately in the space. We visualize these stripes in the AV-MNIST dataset with 3-dimensional embeddings in Fig. 9. Audio (triangles) and vision (squares) embeddings lie in separate regions of the latent space. Nevertheless, they form "semantic stripes" with each other, in which classes (colors) are matched in stripes with the corresponding class of a different modality.

Table 6: Classification results in MultiBench datasets compared with CoMM Dufumier et al. (2025) and with CoMM + Yaras et al. (2025).

| | MOSI | | UR-FUNNY | |
|---|---|---|---|---|
| Method | Gap | Acc | Gap | Acc |
| CoMM Dufumier et al. (2025) | 0.33 | 65.91 | 0.81 | 62.83 |
| CoMM + Yaras et al. (2025) | 0.28 | 66.42 | 0.63 | 61.10 |
| CoMM + Ours | **0.24** | **67.65** | **0.77** | **63.25** |

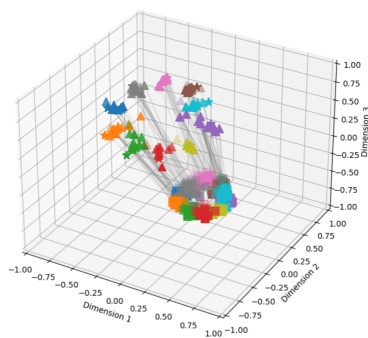

Figure 9: Semantic stripes in the AV-MNIST dataset for CLIP (LT).

**Intra-modal expressiveness.** Modality specific features are essential for some instance level or downstream tasks. Reducing the gap between modalities does not imply discarding or collapsing the unique characteristics of each modality. The proposed losses act only on the relative positioning of positive pairs. They encourage samples that share the same semantics to move closer without limiting the expressiveness of the individual encoders, as confirmed by our experiments. To support such claims, we conduct an additional experiment to verify whether intra-modal discriminative features are preserved. We use pretrained encoders on the MSR-VTT dataset, specifically Eva-CLIP for visual features, BEATs for audio features, and BERT for textual features. After extracting the embeddings, we apply both the k-NN algorithm and a simple linear classifier (In MSR-VTT each sample belongs to one of the 20 categories). Table 7 reports the accuracy obtained by these two methods when applied independently to the features of each modality. Each experiment is conducted 5 times. The results show that the expressive power of each individual encoder remains unaffected by the introduction of our additional losses and by the consequent reduction of the modality gap. In fact, both the kNN and linear probing performance are essentially unchanged in the scenario where the gap is present (CLIP (LT)) and in the scenario where the gap is reduced (Ours). These results mean that the discriminative intra-modality features are preserved even when the gap is mitigated, thus the embeddings are likely to preserve the intra-modality features.

Table 7: Accuracy of kNN and linear classification on modality-specific features (visual, textual, and audio) using pretrained encoders. The results demonstrate that the proposed losses preserve discriminative intra-modality features, as accuracy remains consistent across both the original and reduced modality gap settings.

| Method | Only Visual Features | | Only Textual Features | | Only Audio Features | |
|---|---|---|---|---|---|---|
| | kNN | Acc | kNN | Acc | kNN | Acc |
| CLIP (LT) | 60.39 ± 2.9 | 63.84 ± 1.2 | 45.79 ± 2.4 | 48.49 ± 2.5 | 38.28 ± 2.5 | 42.36 ± 3.5 |
| CLIP (FT) | 59.62 ± 2.3 | 64.40 ± 0.8 | 44.61 ± 3.3 | 47.90 ± 0.5 | 37.71 ± 2.7 | 41.80 ± 3.4 |
| Ours | 60.39 ± 2.1 | 64.20 ± 1.0 | 44.34 ± 2.5 | 49.98 ± 1.8 | 37.50 ± 2.8 | 42.26 ± 2.6 |

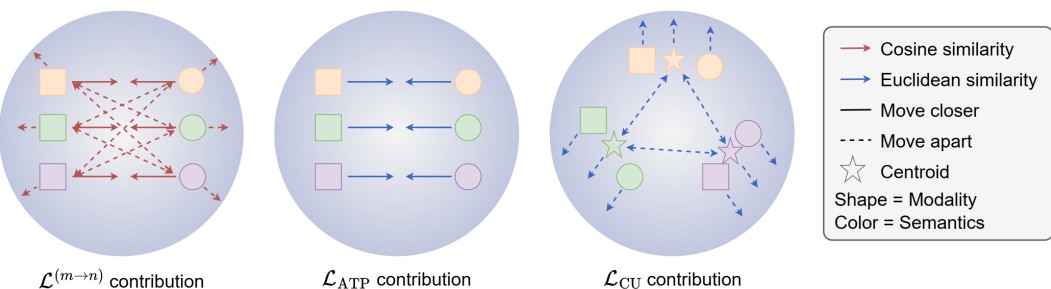

Figure 10: Schematic representation of losses contributions.

## A.4 ABLATION STUDIES

**Why using $\mathcal{L}^{(m \to n)}$ in combination with the proposed losses?** $\mathcal{L}_{\text{ATP}}$ and $\mathcal{L}_{\text{CU}}$ regularise the global geometry (closing the modality gap and preventing centroid collapse), but they do not provide the instance-level repulsion that keeps different samples apart. Instead, the InfoNCE supplies exactly that term, so all three losses are somewhat complementary required to obtain representations that are simultaneously well aligned (no gap), uniformly spread, and discriminative. To prove our claims, we perform an ablation study to highlight the critical role of the InfoNCE loss ($\mathcal{L}^{(m \to n)}$) in achieving optimal performance. While overall semantic alignment can still be attained using a combination of $\mathcal{L}_{ATP}$ and $\mathcal{L}_{CU}$, as evidenced by comparable results in R@5 and R@10, the absence of $\mathcal{L}^{(m \to n)}$ leads to a noticeable drop in R@1 performance. This indicates that the framework becomes less accurate in retrieving the top-1 correct caption when the contrastive loss is removed, underscoring its importance for fine-grained alignment between modalities.

Table 8: Ablation on the contribution of $\mathcal{L}^{(m \to n)}$ in combination with the proposed losses.

| $\mathcal{L}^{(m \to n)}$ | Cos TP ↑ | Gap ↓ | T2V R@1 | T2V R@5 | T2V R@10 | V2T R@1 | V2T R@5 | V2T R@10 |
|---|---|---|---|---|---|---|---|---|
| ✗ | **0.71** | 0.05 | 30.0 | 68.3 | 83.0 | 31.2 | 69.3 | 82.0 |
| ✓ | 0.63 | **0.04** | **35.2** | **71.5** | **84.2** | **36.4** | **72.1** | **83.8** |

Furthermore, we visualize the contributions of each of the three losses $\mathcal{L}^{(m \to n)}$, $\mathcal{L}_{\text{ATP}}$, and $\mathcal{L}_{\text{CU}}$ in Fig. 10. The first one, the InfoNCE loss, aligns multimodal matching pairs while spreading all non-matching ones apart. This dual behavior encourages the formation of a gap between modalities, as demonstrated in numerous previous works Liang et al. (2022); Shi et al. (2023). The InfoNCE loss function often becomes trapped in local minima, where the loss is minimized (semantic alignment is maximized) but a non-zero gap remains Shi et al. (2023). The $\mathcal{L}_{\text{ATP}}$ loss forces the match among true pairs, consistently reducing the modality gap. However, the $\mathcal{L}_{\text{ATP}}$ does not consider non-matching pairs, potentially making the latent space collapse in small regions, thus the contribution of $\mathcal{L}_{\text{CU}}$ becomes crucial. Indeed, the $\mathcal{L}_{\text{CU}}$ loss spreads the semantic centroids (i.e., the centroids of matching pairs) while preserving the alignment.

**Ablation on $\mathcal{L}_{\text{ATP}}$ and $\mathcal{L}_{\text{CU}}$.** We conduct an ablation study for the proposed loss functions. We train from scratch two ResNet50 encoders with OpenCLIP Cherti et al. (2023) on the CIFAR10 dataset for 100 epochs. We set the temperature parameter of the CLIP objective to 0.07 and we perform sensitivity experiments, weighting the two proposed losses. To this purpose, we introduce two new hyperparameters for equation 10, $\lambda_1$ and $\lambda_2$, combining the losses as

$$\mathcal{L}_{\text{CL}_{\text{gap}}} = \mathcal{L}^{(m \to n)} + \lambda_1 \mathcal{L}_{\text{ATP}} + \lambda_2 \mathcal{L}_{\text{CU}}. \qquad (21)$$

Tab. 9 reports the results for retrieval (R@1) and clustering (V) across all the investigated configurations.

It is interesting to note that the configuration with $\lambda_1 = \lambda_2 = 0$ (the standard InfoNCE loss alone) achieves the lowest performance in both retrieval and clustering tasks. Furthermore, the results show a clear trend: increasing $\lambda_2$ consistently leads to improved performance, indicating the importance of

Table 9: Sensitivity study on $\lambda_1$ (row) and $\lambda_2$ (column) showing retrieval (R@1), clustering (V-Measure), and their average (Avg) on the CIFAR10 dataset. Bold indicates the best value; underlines highlight competitive second-best settings.

| $\lambda_1 \backslash \lambda_2$ | 0.00 | 0.25 | 0.50 | 0.75 | 1.00 |
|---|---|---|---|---|---|
| 0.00 | R@1: 79.39
V: 61.65
Avg: 70.52 | R@1: 78.64
V: 63.67
Avg: 71.16 | R@1: 83.61
V: 67.70
Avg: 75.66 | R@1: 84.05
V: 66.87
Avg: 75.46 | R@1: **86.20**
V: 66.64
Avg: 76.42 |
| 0.25 | R@1: 80.59
V: 66.05
Avg: 73.32 | R@1: 79.39
V: 63.26
Avg: 71.33 | R@1: 80.15
V: 64.47
Avg: 72.31 | R@1: 84.06
V: 69.93
Avg: 76.99 | R@1: 83.82
V: 69.77
Avg: 76.80 |
| 0.50 | R@1: 82.37
V: 68.96
Avg: 75.67 | R@1: 79.34
V: 63.16
Avg: 71.25 | R@1: 81.69
V: 66.90
Avg: 74.30 | R@1: 82.43
V: 65.17
Avg: 73.80 | R@1: 84.14
V: 70.42
Avg: 77.28 |
| 0.75 | R@1: 79.23
V: 65.27
Avg: 72.25 | R@1: 78.40
V: 62.89
Avg: 70.65 | R@1: 82.20
V: 67.60
Avg: 74.90 | R@1: 81.82
V: 65.74
Avg: 73.78 | R@1: 84.03
V: 70.17
Avg: 77.10 |
| 1.00 | R@1: 80.80
V: 66.84
Avg: 73.82 | R@1: 78.13
V: 61.55
Avg: 69.84 | R@1: 78.89
V: 60.67
Avg: 69.78 | R@1: 84.21
V: 70.88
Avg: 77.55 | R@1: 84.64
V: **71.47**
Avg: **78.06** |

Table 10: Quantification of collapse in the latent space with (w/) and without (w/o) $\mathcal{L}_{\text{CU}}$.

| | Gap | Cos TP | AV (t) | AV (v) | T2V R@1 | V2T R@1 |
|---|---|---|---|---|---|---|
| w/o $\mathcal{L}_{\text{CU}}$ | 0.08 | 0.76 | 0.122 | 0.091 | 30.86 | 31.64 |
| w/ $\mathcal{L}_{\text{CU}}$ | 0.09 | 0.58 | 0.001 | 0.005 | 37.5 | 38.67 |

the $\mathcal{L}_{\text{CU}}$ component. In contrast, performance appears less sensitive to variations in $\lambda_1$, suggesting a smaller but still complementary contribution of the $\mathcal{L}_{\text{ATP}}$ term. Additionally, without $\mathcal{L}_{\text{CU}}$, the $\mathcal{L}_{\text{ATP}}$ only makes representations latent space collapsing in a small region of the space, as we show in Fig. 11, thus limiting the expressiveness of the space in downstream tasks. Involving both the proposed losses, instead, builds a more compact and well-aligned latent space, as the third plot in Fig. 11 shows.

A quantitative measure of this collapse is also given in Tab. 6, where we report the gap, the cosine true pairs, and the additional Angular Value (AV) per modality. This metric measures the intra-modal average cosine similarity. It indicates how much the embeddings of a single modality are spread across the hypersphere. A value of this metric higher than 0 indicates that all the embeddings are very close to each other, while a value of 0 means that the intra-cosine similarity ranges from -1 to 1, indicating a good sparsification of the latent space. We conduct these experiments on the MSCOCO dataset with two ResNet50 encoder backbones. Table 6 shows that without $\mathcal{L}_{\text{CU}}$ the space built by solely the InfoNCE and $\mathcal{L}_{\text{ATP}}$ losses tends to have true pairs closer (Cos TP higher) at the cost of a lower sparsification (AV higher). This directly impacts retrieval results as the space is much more condensed and it is harder for the model to discriminate between matching and non-matching pairs.

**Promoting centroids uniformity vs overall uniformity.** Wang & Isola (2020) proved and empirically demonstrated that the conventional InfoNCE loss can be decomposed into two objectives, one that pulls together embeddings from similar samples, and another that spreads all embeddings evenly on the hypersphere to avoid collapse and improve generalization. The latter (i.e., the embeddings spread) applies to all embeddings regardless of the class. However, enforcing uniformity at the sample level tends to scatter representations arbitrarily, potentially disrupting the tight alignment that the alignment term aims to enforce. For this reason, we adapt the uniformity principle to the multimodal setting by applying it not to individual embeddings, but to the centroids of aligned samples, that is, to the average representations of semantically matching pairs across modalities. This design retains the benefits of uniform coverage of the latent space while avoiding crucial interference with alignment. The $\mathcal{L}_{\text{CU}}$ loss encourages the centroids of aligned multimodal samples to be well-separated on the hypersphere, effectively ensuring that different semantic concepts remain distinguishable. At the same time, the $\mathcal{L}_{\text{ATP}}$ loss guarantees that all modalities representing the same semantic concept are tightly

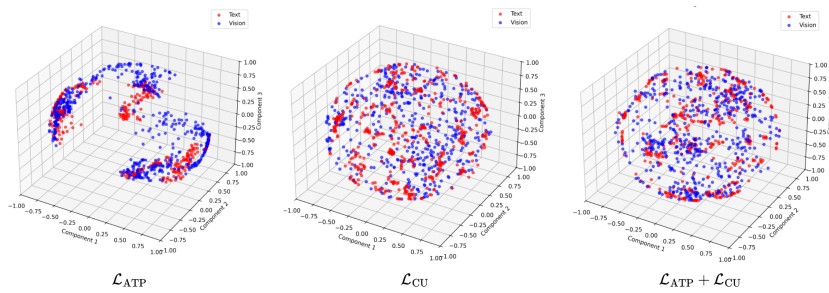

$\mathcal{L}_{\text{ATP}}$ $\qquad$ $\mathcal{L}_{\text{CU}}$ $\qquad$ $\mathcal{L}_{\text{ATP}} + \mathcal{L}_{\text{CU}}$

Figure 11: Latent space visualization of different learned spaces in ablation studies, with the precise effect of each of the proposed losses. Experiments with ResNet50 from scratch on MS COCO dataset.

grouped around their centroid. To prove the limitations of the uniform spread of all embeddings, we perform an ablation study on the CIFAR10 dataset, in which we apply the uniformity loss proposed in Wang & Isola (2020) and our $\mathcal{L}_{\text{CU}}$ loss. As it is clear from Tab. 11, without any other changes, the proposed $\mathcal{L}_{\text{CU}}$ improves the performance of both retrieval (R@1) and clustering (V-Measure) over the uniformity loss of Wang & Isola (2020).

Table 11: Comparison between uniformity and our centroid-based uniformity loss.

| Method | R@1 ↑ | V-Measure ↑ |
|---|---|---|
| $\mathcal{L}_{\text{uniform}}$ Wang & Isola (2020) | 82.47 | 64.72 |
| $\mathcal{L}_{\text{CU}}$ (ours) | **84.64** | **71.47** |

### A.5 LIMITATIONS

While our method effectively reduces the modality gap and enhances group-wise alignment across multiple modalities, it does not directly address potential limitations arising from highly imbalanced datasets or scenarios where semantic alignment is ambiguous or weakly defined. Indeed, we observe in our experiments, which contain datasets with diverse modalities imbalances, a different impact of the proposed method. While MSR-VTT and MSCOCO have one diverse matching sample for each modality (i.e., each image corresponds to a single textual description in MSCOCO and similar in MSR-VTT), so there is no imbalance between modalities, CIFAR10 and AV-MNIST have a different structure. AV-MNIST contains three modalities, where images and audio have the same size, while only 10 diverse textual captions are present in the dataset, making this modality slightly imbalanced. What is more, the CIFAR10 datasets contains only 60k images but only 10 testual captions corresponding to "A photo of class". The CIFAR10 is the most imbalanced dataset in our set of experiments. Coming to our method tests, we can observe that, even though our method outperforms previous ones in all the experiments, the increase in the CIFAR10 is the most tight (+0.9 in V-Measure), followed by the AV-MNIST (+4.4 in V-Measure), likely due to the imbalanced modalities structure. Indeed, we find the highest improvements in the balanced datasets, in which we got +10.65 in V-Measure for MSCOCO and +8.8 in MSR-VTT.

Moreover, although we validate our approach on both bimodal and trimodal benchmarks, extending the evaluation to more complex or underexplored modality combinations remains future work.

