# OpenReview forum: "Closing the Modality Gap Aligns Group-Wise Semantics"
_ICLR.cc/2026/Conference — ICLR 2026 Poster_

### Official Review · Reviewer_pS85 · 2025-10-25

**Soundness:** 3
**Presentation:** 3
**Contribution:** 2
**Rating:** 4
**Confidence:** 4

**Summary:**

This paper proposes an approach to address the modality gap by introducing two complementary loss functions. The first, $\mathcal{L}\_{\text{ATP}}$ directly aligns modalities by minimizing the Euclidean distance between true pairs. The second,$\mathcal{L}\_{\text{CU}}$ makes semantic concepts (represented by their centroids) sparse and uniformly spread throughout the latent space.

These two losses are combined as $\mathcal{L}\_{\text{gap}}$ and added to the standard contrastive loss. This method functions by pulling embeddings of the same concept from different modalities together ($\mathcal{L}\_{\text{ATP}}$) while simultaneously pushing the centroids of different concepts apart ($\mathcal{L}\_{\text{CU}}$).

Then, this leads to reducing modality gap, not only with 2 modality like CLIP, but additional modalities within 3 or 4 modalities, which extend into general settings, improves performance in downstream tasks.

**Strengths:**

- Recent studies have sought to interpret why CLIP-based models inherently exhibit a gap between modalities (e.g., intra-modal misalignment [1]). This paper clearly defines this problem and proposes an approach to mitigate the modality gap by grouping clusters with similar concepts to enhance intra-modal alignment and disentangle their representations.

- In addition, the paper provides both qualitative and quantitative evidence supporting the effectiveness of the proposed method (e.g., qualitative results in Fig. 2 and Fig. 6; quantitative results in Fig. 3 and Fig. 4). The demonstrated reduction of the modality gap also leads to improved performance on downstream tasks, consistent with the paper’s main objectives.

&nbsp;

[1] Mistretta, Marco, et al. "Cross the gap: Exposing the intra-modal misalignment in clip via modality inversion." ICLR 2025\

**Weaknesses:**

**Major**

In my understanding, there remains a fundamental concern about whether reducing the modality gap is always beneficial [1][2]. For instance, in an image–text pair where the image contains additional contextual elements (e.g., trees) not mentioned in the text, the discrepancy reflects complementary rather than misaligned information. Specifically, let consider an image–text pair where the image depicts a dog playing with a frisbee under trees (Image Modality), while the text annotation reads *“The dog is playing with a frisbee.”* (Text Modality). In this case, the image contains additional, unique information (e.g., the presence of trees) that the text does not. This suggests that a certain degree of modality discrepancy can be natural or even desirable, as each modality may capture complementary aspects of the scene [3]. However, the paper assumes that reducing this gap is inherently optimal for multimodal learning but does not provide some justifications for this claim, just relying mainly on partial empirical observations (e.g., Fig. 6; Tables 6 and 7).

[1] Yaras, Can, et al. "Explaining and mitigating the modality gap in contrastive multimodal learning." CPAL 2025\
[2] Schrodi, Simon, et al. "Two effects, one trigger: On the modality gap, object bias, and information imbalance in contrastive vision-language representation learning." ICLR 2025\
[3] Sammani, Fawaz, and Nikos Deligiannis. "Interpreting and analysing CLIP's zero-shot image classification via mutual knowledge." NeurIPS 2024

**Minor**

- The paper appears to address a slightly trivial problem — primarily focusing on reducing the modality gap — without presenting a sufficiently distinctive or theoretically grounded contribution. It seems to be incremental contribution.
- Limited comparison: The paper lacks comprehensive comparisons with other relevant approaches, such as  OTI/OVI [1] and CoMM [2].

[1] Mistretta, Marco, et al. "Cross the gap: Exposing the intra-modal misalignment in clip via modality inversion." ICLR 2025\
[2] Dufumier, Benoit, et al. "What to align in multimodal contrastive learning?." ICLR 2025


(**Note**: *The authors do not need to respond to these weaknesses, but to the questions listed below.*)

**Questions:**

1. (Related to the major weakness) What are the benefits of reducing the modality gap beyond merely improving model performance? While prior works have suggested that narrowing this gap leads to better VLMs or multimodal representations based on empirical results, a deeper discussion is needed. The authors should clarify why reducing the gap is inherently advantageous, rather than preserving some degree of modality-specific uniqueness.
2. The relative strength of the proposed $\mathcal{L}_{\text{gap}}$ compared to existing methods, such as OTI, OVI, or CoMM, remains unclear. Therefore, additional experiments are likely required to demonstrate whether this approach provides a novel or complementary benefit when applied alongside other methods. Comparisons do not need to be limited to OTI, OVI, or CoMM; any reasonable alternative methods may be used.
3. As mentioned in the Limitations (Appendix A.3), the method may struggle when datasets are highly imbalanced across modalities. As a minor concern, it would be valuable to include additional experiments or analyses evaluating performance under varying degrees of modality imbalance (e.g., via controlled imbalance scenarios or dataset subsampling). The choice of datasets for such experiments can be flexible.

=======================================================

**Note**: I acknowledge that I may have partially misunderstood certain aspects of the paper. Therefore, I am willing to raise my rating score if these questions and concerns are adequately addressed.

---

> ### Author Response · Authors · 2025-11-21
>
> **Q1 benefit of reducing the modality gap.** We would like to thank the Reviewer for giving us the opportunity to discuss on this interesting aspect of the work.
> As we pointed out in the introduction of the paper, the reasons for mitigating or untouching such a gap are not well-grounded and the advantages both theoretical and in common downstream tasks such as retrieval are inconsistent throughout the literature.
>
> On the contrary, we claim that, by reducing the modality gap, we can build a latent space where embeddings with the same semantic concept are grouped together, regardless of the modality. Indeed, if we look at a portion of the latent space, we can observe embeddings from different modalities that share the same semantics, while simultaneously preserving modality-specific features. This is possible as the gap is orthogonal to the span of semantic embeddings(Zhang et al., 2023). Thus, even when we close the modality gap, the uniqueness of modality features are preserved. To prove this, we conduct an additional experiment to verify whether intra-modal discriminative features are preserved. We use pretrained encoders on the MSR-VTT dataset, specifically EVA-CLIP for visual features, BEATs for audio features, and BERT for textual features. After extracting the embeddings, we apply both the k-NN algorithm and a simple linear classifier (In MSR-VTT each sample belongs to one of the 20 categories). The table below reports the accuracy obtained by these two methods when applied independently to the features of each modality. Each experiment is conducted 5 times. The results show that the expressive power of each individual encoder remains unaffected by the introduction of our additional losses and by the consequent reduction of the modality gap. In fact, both the k-NN and linear probing performance are essentially unchanged in the scenario where the gap is present (CLIP LT) and in the scenario where the gap is reduced (Ours). These results mean that the discriminative intra-modality features are preserved even when the gap is mitigated, thus the embeddings are likely to preserve the intra-modality features.
>
> | Model | Visual Features (kNN / Acc) | Textual Features (kNN / Acc) | Audio Features (kNN / Acc) |
> | :--- | :---: | :---: | :---: |
> | CLIP (LT) | $60.39 \pm 2.9$ / $63.84 \pm 1.2$ | $45.79 \pm 2.4$ / $48.49 \pm 2.5$ | $38.28 \pm 2.5$ / $42.36 \pm 3.5$ |
> | CLIP (FT) | $59.62 \pm 2.3$ / $64.40 \pm 0.8$ | $44.61 \pm 3.3$ / $47.90 \pm 0.5$ | $37.71 \pm 2.7$ / $41.80 \pm 3.4$ |
> | Ours | $60.39 \pm 2.1$ / $64.20 \pm 1.0$ | $44.34 \pm 2.5$ / $49.98 \pm 1.8$ | $37.50 \pm 2.8$ / $42.26 \pm 2.6$ |
>
> Therefore, reducing the modality gap helps building a more structured latent space with portions of the space that share the same semantics, regardless of the original modality, while also preserving modality-specific features. Reasonably, this directly translates in improved performance in group-wise tasks and preserved performance in instance-wise ones.
> We have revised the paper including this discussion in the Appendix. We would likt to thank the Reviewer once again for the interesting point.

---

> ### Author Response · Authors · 2025-11-21
>
> **Q2 additional experiments.** We would like to thank the Reviewer for the valuable suggestion. In the paper, we have already compared the proposed method with the method proposed in (Yaras et al., 2025) in Tab. 1 and Tab.2 and with NotAGap (Fahim et al., 2024) in Tab. 2. Nevertheless, according to the Reviewer's suggestion, we performed additional comparisons. We rerun the experiments using CoMM repository in three datasets (MOSI, UR-FUNNY, MuSTARD) for the classification task. Then, we apply on CoMM the method proposed in (Yaras et al., 2025) for a further comparison (CoMM + (Yaras et al., 2025)). Finally, we add our proposed losses to the CoMM framework (CoMM + Ours line in the table below). We report gap and accuracy measures. As it is clear from the table below, the proposed loss combination improves results in terms of accuracy with respect to CoMM and reduces the gap in all the datasets.
>
>
> | Method      | MOSI Gap | MOSI Acc | UR-FUNNY Gap | UR-FUNNY Acc |
> |-------------|----------|----------|--------------|--------------|
> | CoMM        | 0.33     | 65.91    | 0.81         | 62.83        |
> | CoMM + (Yaras et al., 2025) | 0.28     | 66.42    | 0.77         | 62.10        |
> | CoMM + Ours | **0.24**     | **67.65**    | **0.74**         | **63.25**        |
>
> Thanks for the valuable suggestion once again, we have added these results to the revised version of the paper.
>
> In (Mistretta et al., 2025), the proposed OTI/OVI are instead introduced for unimodal tasks like image-to-image retrieval and image classification in the two-modal case only, while our paper focuses on cross-modal tasks generalizable from 2 up to more modalities, so we originally did not compare our method against OTI. Nevertheless, we have now conducted an experiment for the image-to-image retrieval task on CIFAR10, comparing CLIP (LT) and CLIP (FT) with OTI and our method, where all the comparisons are based on the same ViT-B16 architecture. As it is clear from the table below, the proposed method outperforms OTI and other methods in this task too.
>
> | Method          | mAP       |
> |-----------------|-----------|
> | CLIP (LT)       | 58.59     |
> | CLIP (LT) + OTI | 30.20     |
> | CLIP (FT)       | 59.12     |
> | CLIP (FT) + OTI | 31.05     |
> | Ours            | **62.21** |

---

> ### Author Response · Authors · 2025-11-21
>
> **Q3 limitations.** We would like to thank the Reviewer for the interesting point. The discussion on the limitations we included in the paper is given by the observed results in our experiments.
> Indeed, our experiments already contains datasets with diverse modalities imbalances. While MSR-VTT and MSCOCO have one diverse matching sample for each modality (i.e., each image corresponds to a single textual description in MSCOCO and similar in MSR-VTT) so there is no imbalance between modalities, CIFAR10 and AV-MNIST have a different structure. AV-MNIST contains three modalities, where images and audio have the same size, while only 10 diverse textual captions are present in the dataset, making this modality slightly imbalanced. What is more, the CIFAR10 datasets contains only 60k images but only 10 testual captions corresponding to "A photo of {class}". The CIFAR10 is the most imbalanced dataset in our set of experiments.
> Coming to our method tests, we can observe that, even though our method outperforms previous ones in all the experiments, the increase in the CIFAR10 is the most tight (+0.9 in V-Measure), followed by the AV-MNIST (+4.4 in V-Measure), likely due to the imbalanced modalities structure. Indeed, we find the highest improvements in the balanced datasets, in which we got +10.65 in V-Measure for MSCOCO and +8.8 in MSR-VTT.
> We have added this discussion in the revised version of the paper to better clarify this point, thanks for the suggestion once again.

---

> ### Comment · Reviewer_pS85 · 2025-11-24
> **Response to the Authors' Rebuttal**
>
> I thank the authors for their detailed response. While the rebuttal has addressed most of my concerns, significant questions remain regarding the modality gap.
>
> I would appreciate a more rigorous theoretical approach or a detailed comparison with the prior works cited in my "Weaknesses" section, specifically [1]. Established literature, notably [1] and [2] ("Mind the Gap"), highlights a counter-intuitive phenomenon where reducing the gap does not necessarily guarantee performance improvements and reducing modality gap sometimes can lead to sub-optimal solutions.
>
> Given this context, could the authors provide a discussion or comparison demonstrating why their proposed approach avoids these pitfalls? If possible, I am looking for detail comparison or theoretical grounding that explains why this method converges to an optimal point, rather than the sub-optimal results warned against in [1] and [2].
>
> &nbsp;
>
> [1] Jiang, Qian, et al. "Understanding and constructing latent modality structures in multi-modal representation learning." CVPR (2023)\
> [2] Liang, Victor Weixin, et al. "Mind the gap: Understanding the modality gap in multi-modal contrastive representation learning." NeurIPS (2022)

---

> > ### Author Response · Authors · 2025-11-25
> >
> > We would like to thank the Reviewer for giving us the opportunity to further discuss on this interesting point.
> > The referenced paper [1] (Jiang et al., 2023) states that "the retrieval performance barely changes when changing the gap between two modalities". Similarly, other papers found that, by reducing the gap, sometimes retrieval/classification results slightly improve [2], sometimes mild positive correlations towards performance decreasing are observed. Overall, there is no clear direction in the literature whether closing the modality gap has a positive impact on retrieval performance. Notably, these papers mostly refer to downstream performance as retrieval performance, not taking group-wise tasks (like clustering) into account when discussing on the implications of reducing the gap for downstream tasks.
> >
> > Throughout our paper, we claim that reducing the modality gap does not severely impact downstream tasks like retrieval, in accordance with [1, among others].
> > Nevertheless, in this paper, we introduce a novel consideration with respect to the literature, which is reducing the modality gap clearly and positively impacts another class of tasks, i.e., group-wise tasks. This is a novel consideration wrt the literature, as we prove that such class of downstream tasks considerably benefit from the gap reduction.
> >
> > Therefore, we confirm the considerations conducted in previous literature on the retrieval performance, while introducing novel considerations on group-wise tasks, which have not been conducted before.
> >
> > Furthermore, both [1] and [2] based their claims mostly on datasets with one major modality (image) and labels that can be translated into text (CIFAR10, StanfordCars, OxfordPets, ...). On the contrary, we test our method in these kind of datasets and also extending it to more large datasets with more modalities and complex interacations, such as MSCOCO and MSR-VTT, making our results robust.
> >
> > Regarding the comparison with the [1] in weaknesses, (Yaras et al., 2025) identifies an intriguing correlation between the modality gap $\Delta$, which they define through the inequality $\Delta \ge 2 \gamma$, or equivalently matching our notation $\Delta \ge 2 \theta$, and the contrastive learning temperature parameter $\tau$. In particular, defining $\beta = 1 / \tau$ and $R = \frac{\partial\theta / \partial t}{\partial \beta / \partial t}$, with $t$ training time, they claim that $R \rightarrow 0$ as $\beta$ increases, that is as the temperature $\tau$ decreases, and thus preventing the modality gap from closing. Since in their analysis they consider a learnable temperature that exponentially decreases during training, they link the modality gap through the training time as
> >
> > \begin{equation}
> > \Delta(t)\ge \Omega(\frac{1}{\log(t)^2}).
> > \end{equation}
> >
> > Therefore, they propose mitigation strategies that operate directly on the temperature parameter avoiding the fast decrease of the temperature $\tau$, which stabilizes the modality gap. One of the most effective methods they report is keeping the temperature fixed at a high value (e.g., 0.1 or 0.07), which they claim help mitigating the modality gap.
> >
> > Following this insight, we also fix the temperature in our framework and introduce dedicated losses to further address the modality gap.
> > That is because, looking at Fig. 3 showing the loss and gradient landscape,
> > the method proposed in (Yaras et al., 2025) corresponding to CLIP (FT) reaches the minimum around 60 degrees, i.e., in the presence of gap, even though partially mitigating the learnable temperature approach CLIP (LT).
> > Conversely, the proposed loss combinations reaches the minimum when the gap is zero, effectively driving the optimization problem toward the gap closure. Moreover, Fig. 3 further shows that when the gap is closer to zero, the contribution to the loss is just matter of the non-matching pairs, therefore strenghtening the alignment. This behavior is proven in Appendix A.1, in which the derivative of our loss $L_{CL_{gap}}$ wrt the shift $\theta = \Delta/2$ is:
> >
> > \begin{equation}
> > \frac{\partial L_{CL_{gap}}}{\partial\theta} = \frac{\partial L_{InfoNCE}}{\partial\theta} + 2\sin\theta,
> > \end{equation}
> >
> > moving the global minimum from $\theta=60°$ of the $L_{InfoNCE}$ to $\theta=0$ adding the term $2\sin\theta$ coming from the derivative of the L_ATP. The detailed proof is in Appendix A.1 of the paper.
> >
> > Finally, in Tab. 1 and 2, we present results and comparisons with Yaras et al. (CLIP (FT)), highlighting that, although adjusting the temperature is indeed beneficial, it is not sufficient on its own to fully mitigate the modality gap phenomenon.
> >
> > We would like to thank the Reviewer once again for the valuable discussion.

---

### Official Review · Reviewer_d752 · 2025-10-29

**Soundness:** 3
**Presentation:** 3
**Contribution:** 3
**Rating:** 6
**Confidence:** 4

**Summary:**

This paper studies the *modality gap* in multimodal contrastive learning — the separation that naturally arises between different modalities (e.g., image and text) in shared embedding spaces. The authors argue that this gap, while having little effect on instance-wise retrieval, negatively impacts *group-wise* semantic alignment such as clustering or class-level organization. To address this, they propose two additional objectives: **Align-True-Pairs** ($L_{\text{ATP}}$) to directly align paired samples, and **Centroid-Uniformity** ($L_{\text{CU}}$) to promote balanced structure across modalities. When combined with InfoNCE, these losses explicitly reduce the modality gap. Experiments on several bi- and tri-modal datasets demonstrate improved clustering performance and maintained retrieval accuracy.

**Strengths:**

1. The paper tackles a timely and underexplored problem in multimodal contrastive learning — the **modality gap** — and offers an insightful analysis of its effects.
2. The distinction between *instance-wise* and *group-wise* consequences of the modality gap is both novel and intuitively well-motivated.
3. The proposed loss functions are simple, differentiable, and easily integrated into existing CLIP-style frameworks.
4. The experimental validation is extensive, spanning multiple bi- and tri-modal datasets and evaluating with diverse metrics such as $V$-Measure, cosine similarity, and retrieval accuracy.
5. The paper is **clearly presented** and well organized, with intuitive figures and a coherent narrative connecting the modality gap, angular separation, and semantic alignment.

**Weaknesses:**

## Major
1. **Limited theoretical grounding.**
   - The main theoretical contribution — that the modality gap influences group-wise semantics — is persuasive but remains largely qualitative. A more formal treatment (e.g., linking it to the alignment–uniformity trade-offs in contrastive theory) would strengthen the argument.
   - The principal practical novelty appears to be Equation (8), as Equation (7) corresponds to the standard “alignment” term introduced by Wang & Isola (2020). Although the proposed term performs well empirically, the paper would benefit from a clearer intuition or justification for it — ideally from an information-theoretic or probabilistic perspective, as is common in recent contrastive learning research.

2. **Incremental methodological contribution.**
   Previous works (e.g., Liang et al., 2022; Schrodi et al., 2025) have already analyzed or attempted to reduce the modality gap. The proposed losses can thus be seen as practical refinements rather than fundamentally new formulations. Nonetheless, emphasizing *group-wise* effects remains an interesting and original perspective.

---

## Minor
1. In Equation (3), the summation index should start at 1, not 0.
2. The meaning of the “semantic class” $c$ in Equation (3) is unclear — the notation suggests a single sample, which makes the formulation ambiguous. It would also be better to avoid reusing the symbol $c$, given that it is later used to denote centroids.
3. Equation (6) should be accompanied by a proof or derivation, ideally included in the appendix.

**Questions:**

1. I would like to clarify a conceptual point: the paper states that the canonical goal of representation learning is to organize the representation space according to semantic structure. I would instead define the canonical goal as *learning representations that capture the task-relevant structure of the data to enable effective downstream learning*. While this may often result in semantic organization, it is not necessarily the goal itself. I’m curious to hear the authors’ perspective on this distinction.
2. Have you analyzed the sensitivity of the results to the weighting of $L_{\text{ATP}}$ and $L_{\text{CU}}$?
3. Could the proposed objectives interact with the temperature $\tau$ in the contrastive loss, or could temperature scaling alone reduce the modality gap?

---

> ### Author Response · Authors · 2025-11-21
>
> **On the theoretical grounding.** We would like to thank the Reviewer for giving us the opportunity to clarify and strengthen our contributions.
> For Equation 8, the theoretical justification can be subsumed from the theoretical framework established by (Wang & Isola, 2020). They demonstrate that the Uniformity of a representation space is best measured by the potential energy of the embeddings under a Gaussian kernel (RBF). Mathematically, minimizing the logarithm of the expected pairwise Gaussian potential (our L_CU) acts as a differentiable proxy for maximizing the entropy of the empirical distribution on the hypersphere. Since the Uniform distribution is the maximum entropy distribution on the hypersphere, minimizing L_CU asymptotically minimizes the divergence between the distribution of semantic centroids and the uniform distribution. Our specific contribution (Eq. 8) is applying this potential energy minimization specifically to the semantic centroids ($c_k$) rather than to the individual modality instances, as the latter approach can disrupt the semantic alignment, as we prove in Tab. 10 in the Appendix.
>
> **On the methodological contribution.** We would like to thank the Reviewer for the possibility to better clarify our contributions. The modality gap is a largely studied phenomenon in multimodal representation learning, with several papers published in the last month. Some papers like (Yaras et al., 2025; Fahim et al., 2025) propose methods to reduce the modality gap and we directly compare our method against them in Tab.2, clearly outperforming both in terms of latent space structure (Gap and Cosine True Pairs). Nevertheless, the modality gap literature does not provide clear justification and direction about whether closing the modality gap is beneficial for downstream tasks as opposite results have been found: as an example, (Yaras et al., 2025) argues that closing the gap may improve the retrieval performance in downstream tasks, while (Schrodi et al., 2025) found that a larger modality gap has a mild positive correlation with downstream performance.
> On the contrary, the main novel contribution of our paper is to provide a justification to close the modality gap that has not been proven in the literature before, to the best of our knowledge: while closing such a gap may not impact instance-wise tasks, it consistently improve group-wise tasks. This finding brings a considerable contribution to the relative literature, clarifying why, when, and how closing the modality gap.
> Together with the main contribution, we also introduce a novel method to mitigate the gap that outperforms previously-proposed methods in every experiment we conduct on diverse benchmarks and tasks.
>
> **On the notation of equation 3.** We would like to thank the Reviewer for pointing this out. We have throughly revised the notation and we clarify that:
> - the centroid $\mathbf{c}^m$ is the centroid of one modality, so it is built as $\mathbf{c}^m = \frac{1}{N} \sum_{i=1}^N \mathbf{z}_i^m$, averaging all the embeddings of a single modality.
> - the semantic centroid for the concept $s$ is represented by $\boldsymbol\mu_s=\frac{1}{2}(\mathbf{z}_s^m + \mathbf{z}_s^n)$, therefore averaging the embeddings from the two modalities $m$ and $n$.
>
> We have revised the paper accordingly and thanks for the suggestion once again.

---

> ### Author Response · Authors · 2025-11-21
>
> **Proof for equation 6.** We would like to thank the Reviewer for this suggestion. We have derived the proof for equation 6 and we have revised the paper accordingly. We provide the proof also here.
>
> For a semantic class $s$ and two modalities $m, n$, the class centroid under a uniform shift $\delta$ is Eq.5:
> \begin{equation}
>     \boldsymbol \mu_s^\delta = \frac{1}{2}\big((\mathbf{z}_s^m+\boldsymbol\delta) + (\mathbf{z}_s^n+\boldsymbol\delta)\big)
>     = \mathbf{\mu}_s^0 + \boldsymbol\delta,
> \end{equation}
>
> where
> \begin{equation}
>     \boldsymbol\mu_s^0 = \frac{1}{2}\left(\mathbf{z}_s^m + \mathbf{z}_s^n\right).
> \end{equation}
>
> We expand the squared deviation:
>
> \begin{equation}
> \left\| \mathbf{z}_s^m - \boldsymbol\mu_s^{\boldsymbol\delta} \right\|_2^2
> = \left\| \mathbf{z}_s^m - (\boldsymbol\mu_s^0 + \boldsymbol\delta) \right\|_2^2 = \left\| (\mathbf{z}_s^m - \boldsymbol\mu_s^0) - \boldsymbol\delta \right\|_2^2
> = \| \mathbf{z}_s^m - \boldsymbol\mu_s^0 \|_2^2 + \|\boldsymbol\delta\|_2^2 - 2\left\langle \mathbf{z}_s^m - \boldsymbol\mu_s^0,\; \boldsymbol\delta \right\rangle.
> \end{equation}
>
> Taking the expectation over classes yields:
>
> \begin{equation}
> \mathbb{E}_s\left[\left\| \mathbf{z}_s^m - \boldsymbol\mu_s^\delta \right\|_2^2\right] = \mathbb{E}_s\left[\left\| \mathbf{z}_s^m - \boldsymbol\mu_s^0 \right\|_2^2\right] + \|\boldsymbol\delta\|_2^2 - 2\,\mathbb{E}_s\left[\left\langle \mathbf{z}_s^m - \boldsymbol\mu_s^0,\; \boldsymbol\delta \right\rangle\right].
> \end{equation}
>
>
> Since $\boldsymbol\delta$ is approximately constant across classes and orthogonal to the span of semantic vectors (Zhang et al., 2023) the cross-term vanishes:
>
> \begin{equation}
> \mathbb{E}_s\left[\left\langle \mathbf{z}_s^m - \boldsymbol\mu_s^0,\; \boldsymbol\delta \right\rangle\right] \approx 0.
> \end{equation}
>
> Thus, we obtain Eq. 6:
> \begin{equation}
> \mathbb{E}_s\left[\left\| \mathbf{z}_s^m - \boldsymbol\mu_s^{\boldsymbol\delta} \right\|_2^2\right]
> \approx
> \mathbb{E}_s\left[\left\| \mathbf{z}_s^m - \boldsymbol\mu_s^0 \right\|_2^2\right] + \|\boldsymbol\delta\|_2^2.
> \end{equation}
>
> **On the goal of representation learning.** We would like to thank the Reviewer for the interesting perspective. The idea of organizing the latent space accordingly to the downstream task would certainly be effective for the given specific downstream task and likely to produce some sort of semantic alignment. Nevertheless, the purpose of pretrained multimodal models, such as ImageBind (Girdhar et al., 2023) and LanguageBind (Zhu et al., 2024), among many others, is to build a semantically meaningful latent space that can be then leveraged to solve any other downstream task. This makes the models generalizable to any required task without needing a retrain for each specific task, which may be highly demanding.
> Within this scope, the method we propose in this paper further improves the generalizability of multimodal models, as it leaves barely unaffected the performance on instance-wise tasks, while considerably improving the performance in group-wise tasks in which methods with modality gap severely fail.
> We would like to thank the Reviewer once again for the interesting point raised.
>
>
> **On the sensitivity of weighting ATP and CU loss.** Thanks for giving us the opportunity to further analyze this point. In Tab.9, we report the results for the sensitivity analysis weighting the ATP and the CU loss. It is interesting to note that the configuration with $\lambda_1 = \lambda_2 = 0$ (the standard InfoNCE loss alone) achieves the lowest performance in both retrieval and clustering tasks. Furthermore, the results show a clear trend: increasing $\lambda_2$ consistently leads to improved performance, indicating the importance of the L_CU component. In contrast, performance appears less sensitive to variations in $\lambda_1$, suggesting a smaller but still complementary contribution of the L_ATP term. Additionally, without L_CU, the L_ATP only makes representations latent space collapsing in a small region of the space, as we show in Fig. 11, thus limiting the expressiveness of the space in downstream tasks. Involving both the proposed losses, instead, builds a more compact and well-aligned latent space, as the third plot in Fig. 11 shows.

---

> ### Author Response · Authors · 2025-11-21
>
> **On the temperature effect.** We thank the Reviewer for giving us the opportunity to go deeper in this point. As noted also in (Yaras et al., 2025), the temperature impacts the gap, therefore we perform an experiment comparing the proposed method and diverse temperature scaling in Fig. 7. We test a set of widely-used temperatures in contrastive learning, [0.01, 0.05, 0.07, 0.1], with the largest temperatures 0.07 and 0.1 reducing the gap with respect to the smaller ones. Nevertheless, the proposed method (with fixed temperature at 0.07) has a larger impact on the gap with respect to the one brought by temperature scaling, reducing the gap more than any other configuration (right-side of Fig. 7). In terms of performance, this directly translates into a higher V-Measure with respect to all other compatisons (left-side of Fig. 7). We have better clarified this aspect in the revised manuscript, thanks for the interesting question!

---

> > ### Comment · Reviewer_d752 · 2025-11-25
> >
> > I thank the authors for their detailed response and for incorporating most of the proposed changes into the manuscript.
> >
> > After carefully reviewing the rebuttal and the revision, I have decided to maintain my original score. I base this decision on two main factors:
> >
> > 1. **Theoretical Grounding:** I do not consider that the theoretical grounding is properly reflected in the paper. Specifically, the statement that “the radial basis function (RBF) kernel in equation 8 is well related to the uniform distribution on the unit hypersphere where multimodal representations lie” does not reflect the level of grounding expected for a spotlight paper at this venue.
> >
> > 2. **Significance of Contribution:** While the core finding—that the importance of the modality gap depends on whether the downstream task is instance-wise or group-wise—is certainly interesting, I do not believe the impact of this contribution is substantial enough to warrant a higher score.
> >
> > **Minor Suggestion:** I strongly recommend the authors rethink the first sentence of the paper. It is the first thing a reader sees, and many (myself included) may disagree with it. The canonical goal of representation learning is not “to embed semantically similar data points nearby and dissimilar ones far apart”, but to learn representations that contain information for a downstream task (or tasks, as you mentioned in your comment) and that this information is accessible by a simple model, e.g., a linear layer. Since this specific definition is not strictly related to the paper's core contribution, you might consider avoiding this sentence to prevent immediate contention.

---

> > > ### Author Response · Authors · 2025-11-26
> > >
> > > We thank the Reviewer for reading our rebuttal and for the minor suggestions.
> > > We have revised the paper with the suggested sentence in Minor Suggestion. The Reviewer can check this update in the revised paper, thank you for the suggestion once again.
> > >
> > > **On the thoretical grounding of the RBF kernel.** The uniformity on the hypersphere of the RBF Gaussian kernel has been already extensively discussed and proven (see Appendix of [1]) in (Wang & isola, ICML 2020). In detail, the authors write: "We want the uniformity metric to be both asymptotically correct (i.e., the distribution optimizing this metric should converge to uniform distribution) and empirically reasonable with finite number of points. To this end, we consider the Gaussian potential kernel (also known as the Radial Basis Function (RBF) kernel)" and that "The average pairwise Gaussian potential is nicely tied with the uniform distribution on the unit hypersphere" [1].
> > >
> > > Nevertheless, to clarify the theoretical grounding of our work and the choice of the RBK kernel, we prove it here once again.
> > >
> > > The Gaussian (RBF) kernel relies on Euclidean distance. Given that we are working on the unit hypersphere (i.e., $\|x\| = \|y\| = 1$) like CLIP, the RBF kernel mathematically transforms into a dot-product kernel, specifically the von Mises-Fisher kernel [2].
> > >
> > > Theoretically, the RBF kernel is defined as:$$K_{\text{RBF}}(x, y) = \exp\left(-\gamma \|x - y\|^2\right)$$
> > >
> > > On the unit hypersphere ($S^{d-1}$), the squared Euclidean distance can be expanded:
> > > $$\|x - y\|^2 = \|x\|^2 + \|y\|^2 - 2x^T y$$
> > >
> > > Since $\|x\| = \|y\| = 1$ by construction, this simplifies to:
> > > $$\|x - y\|^2 = 2 - 2x^T y$$
> > >
> > > Substituting this back into the RBF kernel:
> > > $$K_{\text{RBF}}(x, y) = \exp\left(-\gamma(2 - 2x^T y)\right) = \exp(-2\gamma) \cdot \exp(2\gamma x^T y)$$
> > >
> > > This results in:
> > >
> > > $$K_{\text{RBF}}(x, y) \propto \exp(\kappa x^T y)$$
> > >
> > > This function, $\exp(\kappa x^T y)$, is exactly the unnormalized density of the von Mises-Fisher (vMF) distribution, which corresponds to the Gaussian distribution of the unit hypersphere, as proven in several works, e.g., [1, 2, 3, 4, 5], among others.
> > >
> > > Specifically, as we set $\kappa$ low, the following holds in our case too:
> > > - "For low values of $\kappa$ the distribution is uniform, i.e. the samples appear as to be uniformly distributed on the sphere". [3]
> > > - "In the limiting case $\kappa \rightarrow 0$, the distribution converges to the uniform distribution over the sphere." [4]
> > > - "when $\kappa = 0$, $f(x|\mu,\kappa)$ (i.e., the distribution) reduces to the uniform density on $S^{d−1}$" (i.e., the hypersphere). [5]
> > >
> > >
> > > [1] T. Wang & P. Isola, "Understanding Contrastive Representation Learning through Alignment and Uniformity on the Hypersphere", ICML 2020.
> > >
> > > [2] K. V. Mardia & P. E. Jupp, Directional Statistics, 1999.
> > >
> > > [3] M. Hasnat, et al., "von mises-fisher mixture model-based deep learning: Application to face verification", arXiv preprint arXiv:1706.04264, 2017.
> > >
> > > [4] K. You, D. Shung, M. Giuffrè, "Learning over von Mises–Fisher Distributions via a Wasserstein-like Geometry", arXiv preprint arXiv:2504.14164v1, 2025.
> > >
> > > [5] A. Banerjee, et al., "Clustering on the Unit Hypersphere using von Mises-Fisher Distributions", Journal of Machine Learning Research, 2005.
> > >
> > > **On the significance of contribution.** We thank the Reviewer for recognizing that our paper has an interesting contribution. Nevertheless, we believe that our contribution on the impact of modality gap that depends on whether the downstream task is instance-wise or group-wise is highly relevant. Moreover, we believe that our contributions are even more significant and go beyond the one highlighted by the Reviewer. Indeed, in this paper we provide a definite answer to the everlasting question in the related literature on why and in which cases closing the modality gap, providing a paradigm shift. This is a significantly novel insight that can consistently contribute to the understanding of the modality gap and of its implications in future multimodal models.
> > > Additionally, previous literature focuses on text-image pairs and on two-modal cases in general (Schrodi et al., 2025; Mistretta et al., 2025; Liang et al., 2022; Shi et al., 2023; among others), without advancing clues on the existance or impact of the modality gap in the case of more modalities. In this paper, we bring the novel insights on the modality gap in the case of more modalities, thus filling a gap in existing literature and significantly contributing to the advancement of the modality gap understanding.

---

### Official Review · Reviewer_h2Ko · 2025-11-01

**Soundness:** 3
**Presentation:** 3
**Contribution:** 2
**Rating:** 6
**Confidence:** 5

**Summary:**

This paper reviews the modality gap in multimodal representation learning, providing insights into the impact of this phenomenon on downstream tasks. By a series of experiments, this paper observes that the modality gap has a limited influence on instance-level tasks such as retrieval. However, it would significantly affect group-wise tasks like clustering. To this end, this paper proposes a new objective function to explicitly align true pairs while promoting latent space uniformity.

**Strengths:**

1. The paper clearly revisited an important distinction between instance-wise and group-wise tasks in multimodal learning, providing a new perspective on the modality gap phenomenon.
2. The theoretical analysis offers a reasonable explanation for why the modality gap affects clustering performance but not retrieval, with the derivation showing how the gap inflates within-class scatter.
3. The paper provides thorough experimental validation across diverse datasets and modalities, with clear visualizations of the latent space before and after closing the gap.

**Weaknesses:**

1. The core idea of combining alignment and uniformity losses is not fundamentally new. Some methods previously established that contrastive learning involves alignment and uniformity, and explored closing the modality gap with similar objectives. The specific combination of LATP and LCU doesn't represent a significant methodological advance.
2. The paper states that "closing the modality gap" is necessary for group-wise tasks, but the marginal improvements on some datasets (like CIFAR-10) suggest the gap may not be the primary limiting factor in many cases.
3. While the paper claims to achieve "nearly zero gap," it doesn't address whether completely closing the gap might lead to other issues like reduced discriminative power between different concepts or increased vulnerability to modality-specific noise.
4. This paper mentions that "the entire latent space collapses into small portions" without the LCU loss, but the ablation studies don't clearly quantify this collapse. The authors should provide visualization or metrics showing the extent of collapse when using only LATP versus the full approach.
5. The paper claims that "closing the modality gap" is the key to improving group-wise tasks. However, could the observed improvements be due to other factors, such as the increased expressiveness of the latent space from your loss formulation rather than specifically from closing the gap?

**Questions:**

See Weaknesses

---

> ### Author Response · Authors · 2025-11-21
>
> **W1 On the ATP and CU loss.** We would like to thank the Reviewer for the opportunity to clarify this point. Although similar losses have been proposed in the literature, we propose a method that is fundamentally new. Indeed, previous methods like (Wang & Isola, 2020) and (Fahim et al., 2024) involve a uniformity loss, which is consistently different from our centroid uniformity. Indeed, their uniformity applies to all embeddings regardless of the class or of the matching instances, scattering representations arbitrarily, potentially disrupting the tight alignment that the alignment term aims to enforce. For this reason, we adapt the uniformity principle to the multimodal setting by applying it not to individual embeddings, but to the centroids of aligned samples, that is, to the average representations of semantically matching pairs across modalities. This design retains the benefits of uniform coverage of the latent space while avoiding crucial interference with alignment. The CU loss encourages the centroids of aligned multimodal samples to be well-separated on the hypersphere, effectively ensuring that different semantic concepts remain distinguishable. At the same time, the ATP loss guarantees that all modalities representing the same semantic concept are tightly grouped together. To prove the limitations of the uniform spread of all embeddings in previous methods, we perform an ablation study on the CIFAR10 dataset, in which we apply the uniformity loss previously proposed and our CU loss. As it is clear from the table below, without any other changes, the proposed $L_{CU}$ improves the performance of both retrieval (R@1) and clustering (V-Measure) over the previously proposed uniformity loss.
>
> **Ablation on uniformity vs proposed centroid uniformity:**
> | Method                   | R@1       | V-Measure |
> |--------------------------|-----------|-----------|
> | L_CL+L_ATP+L_uniform (Wang&Isola, 2020) | 82.47     | 64.72     |
> | L_CL+L_ATP+L_CU (Ours)   | **84.64** | **71.47** |
>
> Therefore, the proposed centroid uniformity loss is different from previous approaches and also way more effective. We report this ablation study also in Tab. 10 of the paper.
>
> **W2 On the improvements.** We would like to thank the Reviewer for the opportunity to clarify our results. In our experiments, the improvements are marginal only in the CIFAR10 dataset, as: on MSCOCO we get +10.65, on AV-MNIST we get +5.1, and on MSR-VTT we get +8.8 in V-Measure. Therefore, our improvements are definitely considerable and consistent. The slight improvement in CIFAR10 is probably due to the dataset itself, as it contains images and a set of 10 labels instead of multiple textual descriptions (as MSCOCO or MSR-VTT, for examples). This makes the clustering task quite easier to perform also in the presence of the modality gap.

---

> ### Author Response · Authors · 2025-11-21
>
> **W3 on the nearly zero gap.** We thank the Reviewer for raising this point, which gives us the opportunity to clarify our claim. We agree that modality specific features can be essential for certain instance level or downstream tasks. Indeed, our method does not aim to remove or weaken such modality specific information. The goal is to reduce the gap between modalities which keeps semantically matching samples far apart in the shared latent space.
> Nevertheless, reducing this gap does not imply discarding or collapsing the unique characteristics of each modality. The proposed losses act only on the relative positioning of positive pairs. They encourage samples that share the same semantics to move closer without limiting the expressiveness of the individual encoders, as confirmed by our experiments. Indeed, in the instance level retrieval task, which is highly sensitive to modality specific cues, performance remain stable or slightly improve, showing that expressive modality dependent information is preserved. At the same time, the global semantic structure of the latent space becomes more coherent, which leads to the observed improvements in group wise tasks.
>
> To support such claims, we conduct an additional experiment to verify whether intra-modal discriminative features are preserved. We use pretrained encoders on the MSR-VTT dataset, specifically EVA-CLIP for visual features, BEATs for audio features, and BERT for textual features. After extracting the embeddings, we apply both the k-NN algorithm and a simple linear classifier (In MSR-VTT each sample belongs to one of the 20 categories). The table below reports the accuracy obtained by these two methods when applied independently to the features of each modality. Each experiment is conducted 5 times. The results show that the expressive power of each individual encoder remains unaffected by the introduction of our additional losses and by the consequent reduction of the modality gap. In fact, both the k-NN and linear probing performance are essentially unchanged in the scenario where the gap is present (CLIP LT) and in the scenario where the gap is reduced (OURS). These results mean that the discriminative intra-modality features are preserved even when the gap is mitigated, thus the embeddings are likely to preserve the intra-modality features.
>
> | Model | Visual Features (kNN / Acc) | Textual Features (kNN / Acc) | Audio Features (kNN / Acc) |
> | :--- | :---: | :---: | :---: |
> | CLIP (LT) | $60.39 \pm 2.9$ / $63.84 \pm 1.2$ | $45.79 \pm 2.4$ / $48.49 \pm 2.5$ | $38.28 \pm 2.5$ / $42.36 \pm 3.5$ |
> | CLIP (FT) | $59.62 \pm 2.3$ / $64.40 \pm 0.8$ | $44.61 \pm 3.3$ / $47.90 \pm 0.5$ | $37.71 \pm 2.7$ / $41.80 \pm 3.4$ |
> | Ours | $60.39 \pm 2.1$ / $64.20 \pm 1.0$ | $44.34 \pm 2.5$ / $49.98 \pm 1.8$ | $37.50 \pm 2.8$ / $42.26 \pm 2.6$ |
>
> We have revised the paper including this interesting discussion and the results according to the Reviewer's suggestion.

---

> ### Author Response · Authors · 2025-11-21
>
> **W4 On the ATP collapse.** We would like to thank the Reviewer for the valuable suggestion. According to it, we perform a quantitative analysis to measure this collapse. In the table below we report the gap, the cosine true pairs, and the additional Angular Value (AV) per modality. This metric measures the intra-modal average cosine similarity. It indicates how much the embeddings of a single modality are spread across the hypersphere. A value of this metric higher than 0 indicates that all the embeddings are very close to each other, while a value of 0 means that the intra-cosine similarity ranges from -1 to 1, indicating a good sparsification of the latent space. We conduct these experiments on the MSCOCO dataset with two ResNet50 encoder backbones for 30k iterations. The table shows that without L_CU the space built by solely the InfoNCE and L_ATP losses tends to have true pairs closer (Cos TP higher) at the cost of a lower sparsification (AV higher). This directly impacts retrieval results as the space is much more condensed and it is harder for the model to discriminate between matching and non-matching pairs. What is more interesting, at the beginning of training the AV is high without the L_CU, then the InfoNCE loss rebalances the latent space and start pushing away non-matching pairs slightly sparsifying the space. On the contrary, when using the L_CU the space is well sparse from the earlier stage of the training.
>
> | Method | Gap | Cos TP | AV (t) | AV (v) | T2V R@1 | V2T R@1 | AV (t) 5k iter | AV (v) 5k iter | AV (t) 15k iter | AV (v) 15k iter |
> | :--- | :---: | :---: | :---: | :---: | :---: | :---: | :---: | :---: | :---: | :---: |
> | w/o L_CU | 0.08 | 0.76 | 0.122 | 0.091 | 30.86 | 31.64 | 0.49 | 0.45 | 0.23 | 0.21 |
> | w/ L_CU | 0.09 | 0.58 | 0.001 | 0.005 | 37.50 | 38.67 | 0.01 | 0.03 | 0.00 | 0.03 |
>
> We have revised the paper including this discussion and the results in the updated paper, we would like to thank the Reviewer once again for the valuable suggestion.
>
> **W5 on the improvements in group-wise tasks.** We would like to thank the Reviewer for giving us the opportunity to clarify this point. The demonstration that the improved performance are due to the gap reduction can be taken from the second baseline following (Yaras et al., 2025), CLIP (FT). In this method, we fix the temperature, which reduced the gap (even though not as much as our method). By slightly reducing the gap with CLIP (FT), also the clustering performance increases. Going ahead, we further reduce the gap with respect to CLIP (FT), with a consequent additional performance improvement. Therefore, we can conclude that the improved clustering performance are likely due to the gap reduction.

---

### Official Review · Reviewer_4NhU · 2025-11-01

**Soundness:** 3
**Presentation:** 2
**Contribution:** 2
**Rating:** 4
**Confidence:** 4

**Summary:**

The authors observe that modality gap does not necessarily affect fine-grained tasks like retrieval in some cases but could affect group-level tasks like clustering in multimodal learning. Based on this, they design a loss function which encourages groups of positive samples to be compactly embedded, while uniformly dispersing negative samples on the hypersphere. They evaluated their approach on several instance-wise and group-wise tasks using standard datasets.

**Strengths:**

1. The observation that modality gap does not necessarily affect fine-grained tasks like retrieval in some cases but could affect group-level tasks like clustering is a novel contribution.

2. The idea of applying cluster-based contrastive losses to close the modality gap in multimodal models is an interesting.

3. The motivation of the paper is clear and the authors do a good job conveying intuitions for some of their claims.

4. Indeed as the authors claim, their empirical results confirm that reducing the modality gap improves group-level tasks like clustering, while leaving tasks like retrieval relatively unaffected in the cases considered.

**Weaknesses:**

1. In Section 3.2, the authors mention that one of the reasons contributing to modality gap is the use of networks with different random initialization for different modalities? How about the scenario where weights are shared across modalities (all modalities are encoded via the same network), since that is the way several widely used multimodal models are trained? Furthermore, although the authors claim this to be one of the sources of modality gap, their solution to resolving it only revolves around modified loss functions, and no attempt to align initialization weights across encoders.

2. The way Section 3.2 is currently organized, it seems as if different random initializations for different modalities is the only source of modality gap. The point about the gap arising from the limitations of the CLIP loss does seems to be made suddenly in the second paragraph and does not follow naturally from the first. I would suggest reorganizing it so as to first enumerate the sources of modality gap, followed by descriptions of how they might be related / affect each other etc., if such relationships exist.

3. In the second paragraph of Section 3.2, authors claim the existence of what they call "semantic stripes" in the representation space of CLIP. Although I can intuitively understand what they are, I would still like to know if the authors have any concrete evidence, theoretical or empirical, to establish their existence. Without any such evidence, such claims seem rather arbitrary.

4. The idea of performing contrastive learning by contrasting cluster centroids is not particularly novel, as research along similar lines is being conducted for several years in the self-supervised learning literature [d, e].

5. It is necessary to investigate further the purpose and effect of ATP. If I understand correctly, ATP is meant to encourage the formation of clusters and prevent positive pairs from being far apart. However, this is not stated explicitly, and there is no term in ATP, unlike the CU loss, involving cluster centroids.

6. It is not entirely true that eliminating modality specific features is guaranteed to leave instance-level tasks like retrieval unaffected. There are several works in the literature [f, g, h] that point to the contrary based on the fact that in many retrieval tasks, there are modality-specific features which could be informative about the downstream task and the instances being compared. The assumption may hold for the experimental settings considered here, but is certainly not generally true.

References:\
[a] Ngiam et al., "Multimodal Deep Learning", ICML 2011.\
[b] Hu et al., "Towards Unsupervised Sketch-based Image Retrieval", BMVC 2022.\
[c] Rastegar et al., "MDL-CW: A Multimodal Deep Learning Framework with Cross Weights", CVPR 2016.\
[d] Caron et al., "Unsupervised Learning of Visual Features by Contrasting Cluster Assignments", NeurIPS 2020.\
[e] Caron et al., "Deep Clustering for Unsupervised Learning of Visual Features", ECCV 2018.\
[f] Jing et al., "Cross-Modal Center Loss for 3D Cross-Modal Retrieval", CVPR 2021.\
[g] Chaudhuri et al., "Cross-Modal Fusion Distillation for Fine-Grained Sketch-Based Image Retrieval", BMVC 2022.\
[h] Ren et al. "Cross-modal retrieval based on multi-dimensional feature fusion hashing", Frontiers in Physics 2024.

Minors:\
Line 142: "in narrow different cones" -> "in different narrow cones"\
Line 207: "suppose to retrieve a cat image caption. The sufficient condition..." -> "suppose to retrieve a cat image caption, the sufficient condition..."

**Questions:**

Please refer to the Weaknesses section.

---

> ### Author Response · Authors · 2025-11-21
>
> We would like to thank the Reviewer for giving us the opportunity to clarify the raised points. Based on these comments, we realized that some parts could be made clearer. As a key general answer, we would like to clarify that although the modality gap exists at initialization, such gap is then preserved (or recreated) by the contrastive behavior, so we directly address this cause of modality gap creation. According to the Reviewer's suggestion, we have reorganized Section 3.2 to better clarify the sources of the modality gap. We go into the details of each raised question/weakness more in detail.
>
> **On sharing the encoder weights.** We would like to thank the Reviewer for the interesting question. Our paper is framed into the literature in which different encoders are used to encode diverse modalities as in, for example, ImageBind (Girdhar et al., 2023) and LanguageBind (Zhu et al., 2024), to cite a few among many others. In this context, previous literature on the modality gap explores the case where different encoders encode diverse modalities (Liang et al., 2022; Shi et al., 2023; Eslami & De Melo, 2025; Schrodi et al., 2025, among others). We follow the same literature branch.
> Nevertheless, we perform an additional experiment involving a model with a shared encoder between the modalities, CoMM (Dufumier et al., 2025), to investigate the modality gap phenomenon in this scenario too. We run experiments with CoMM and then with CoMM together with our loss functions in two different datasets MOSI and UR-FUNNY for the classification task from the MultiBench benchmark.
> Results are shown in the following table.
>
>
> | Method      | MOSI Gap | MOSI Acc | UR-FUNNY Gap | UR-FUNNY Acc |
> |-------------|----------|----------|--------------|--------------|
> | CoMM        | 0.33     | 65.91    | 0.81         | 62.83        |
> | CoMM + Ours | **0.24**     | **67.65**    | **0.77**         | **63.25**        |
>
> The table shows that the modality gap exists even in this case, when a single encoder is used for two modalities. Moreover and once again, the proposed method reduces the modality gap leading to improved performance in group-wise tasks like classification.
>
> We have added this additional experiment and this discussion in the revised manuscript, thanks again for the valuable comment.
>
> **On the gap at initialization.** We thank the Reviewer for giving us the opportunity to clarify this section. According to (Liang et al., 2022) that first discovered the modality gap phenomenon, the gap exists at initialization and it is then preserved by the conventional contrastive loss. Nevertheless, the key reason for the modality gap is the contrastive behavior of the InfoNCE loss and not the initialization. To prove it, we compare two configurations with the same setup, model (ResNet50), and loss (InfoNCE). We enforce an initial sparsification of the learned space for the first epoch so that the encoders first learn to sparsify the embeddings, regardless of the modality. We do so by using as loss the uniformity loss over the hypersphere by (Wang & Isola, 2020).
> However, once the encoders learn the sparsification and we activate instead the conventional contrastive learning loss, the encoders start to separate the modalities again and the gap begins to be recreated, as we show in Fig. 8 of the revised paper. We conduct the same experiment with the proposed losses to close the gap, and no effect is revealed with the initial sparsification as well. Therefore, although the gap is created at initialization, this does not impact the learning procedure and it is therefore the conventional contrastive loss function that tends to create such a gap.
> We have revised the manuscript adding this discussion, together with some plots proving our claims in the Appendix of the revised version, thanks for the comment once again.
>
> **On the semantic stripes.** Thanks for raising this point that can help clarifying our paper.
> What we named "semantic stripes" is the local minimum in which contrastive methods fall due to the contrastive dynamics. Such behavior is evident in (Shi et al., 2023) Fig.1 and Fig.2, and also in (Liang et al., 2022) Fig.1. Furthermore, we revise the paper by adding a figure that shows such semantic stripes. The Reviewer can find such representation in Figure 9 of the revised paper, in which it is clear how embeddings form semantic stripes being separated by the modality gap.
> We would like to thank the Reviewer for the valuable comment once again.

---

> ### Author Response · Authors · 2025-11-21
>
> **On the ATP loss.** Thanks for giving us the opportunity to clarify this point. The ATP loss does not involve centroids, and its purpose is not to encourage the formation of clusters, but rather to couple positive matching pairs instance-wisely. Instances centroids are only considered in the CU loss, as the Reviewer correctly pointed out.
>
> **On eliminating modality-specific features.** We thank the Reviewer for raising this point, which gives us the opportunity to clarify our claim. We agree that modality specific features can be essential for certain instance level or downstream tasks. Indeed, our method does not aim to remove such modality specific information. The goal is to reduce the gap between modalities which keeps semantically matching samples far apart in the shared latent space.
> Nevertheless, reducing this gap does not imply discarding or collapsing the unique characteristics of each modality. The proposed losses act only on the relative positioning of positive pairs. They encourage samples that share the same semantics to move closer without limiting the expressiveness of the individual encoders, as confirmed by our experiments. Indeed, in the instance level retrieval task, which is highly sensitive to modality specific cues, performance remains stable or slightly improves, showing that expressive modality dependent information is preserved. At the same time, the global semantic structure of the latent space becomes more coherent, which leads to the observed improvements in group wise tasks.
>
> To support such claims, we conduct an additional experiment to verify whether intra-modal discriminative features are preserved. We use pretrained encoders on the MSR-VTT dataset, specifically EVA-CLIP for visual features, BEATs for audio features, and BERT for textual features. After extracting the embeddings, we apply both the k-NN algorithm and a simple linear classifier (In MSR-VTT each sample belongs to one of the 20 categories). The table below reports the accuracy obtained by these two methods when applied independently to the features of each modality. Each experiment is conducted 5 times. The results show that the expressive power of each individual encoder remains unaffected by the introduction of our additional losses and by the consequent reduction of the modality gap. In fact, both the k-NN and linear probing performance are essentially unchanged in the scenario where the gap is present (CLIP LT) and in the scenario where the gap is reduced (Ours). These results mean that the discriminative intra-modality features are preserved even when the gap is mitigated, thus the embeddings are likely to preserve the intra-modality features.
>
> | Model | Visual Features (kNN / Acc) | Textual Features (kNN / Acc) | Audio Features (kNN / Acc) |
> | :--- | :---: | :---: | :---: |
> | CLIP (LT) | $60.39 \pm 2.9$ / $63.84 \pm 1.2$ | $45.79 \pm 2.4$ / $48.49 \pm 2.5$ | $38.28 \pm 2.5$ / $42.36 \pm 3.5$ |
> | CLIP (FT) | $59.62 \pm 2.3$ / $64.40 \pm 0.8$ | $44.61 \pm 3.3$ / $47.90 \pm 0.5$ | $37.71 \pm 2.7$ / $41.80 \pm 3.4$ |
> | Ours | $60.39 \pm 2.1$ / $64.20 \pm 1.0$ | $44.34 \pm 2.5$ / $49.98 \pm 1.8$ | $37.50 \pm 2.8$ / $42.26 \pm 2.6$ |
>
> We have revised the paper including this interesting discussion and the results according to the Reviewer's suggestion.
>
> **Minors:** Thanks for pointing them out, we have corrected the paper according to the Reviewer's suggestion.

---

> > ### Comment · Reviewer_4NhU · 2025-11-24
> >
> > I thank the authors for taking the time to respond to my comments and provide several experiments and analyses to this end. A number of my concerns have been addressed, however, some still remain, which I state below:
> >
> > 1. If the uniformity loss over the hypersphere to encourage sparsification can directly reduce the modality gap, then why not just use that? What is the added advantage the proposed method presents over the uniformity loss? Also, the results in Figure 8 are obtained by first sparsifying and then applying the contrastive loss. While the presented results are certainly helpful and adds clarity to the paper, since contrastive loss is known to preserve / recreate the modality gap as noted by the authors, applying it at the last stage should unsurprisingly lead to the observed results. Therefore, I would be curious to know what happens when the two are optimized simultaneously. My intuition is that if a proper balance is struck, then a right level of contrastive semantics alongside desired sparsification can be achieved.
> >
> > 2. Is there any theoretical reason why the proposed loss would not eliminate modality-specific information? Apparently, there does not seem to be any component in its formulation that explicitly retains them. The authors claim that the proposed loss only affects the relative positioning of the positive pairs. However, even in this process, modality-specific features could be eliminated, as pointed out in [f, g, h]. Roughly speaking, the reason for this is that a model optimizing to align positive pairs converges to a shared representation, eliminating everything else that is not shared. Therefore, some special handling is needed to prevent the loss of information that is not shared, but could still be useful for a downstream task. In the proposed method, if there is any implicit mechanism through which this becomes possible leading to the corresponding empirical observations presented by the authors, a clarification on the same is needed.

---

> > > ### Author Response · Authors · 2025-11-24
> > >
> > > We would like to thank the Reviewer for carefully reading our rebuttal.
> > >
> > > 1. Regarding the uniformity loss, enforcing uniformity at the sample level as done in (Wang & Isola, 2020) tends to scatter representations arbitrarily, potentially disrupting the tight alignment that the alignment term aims to enforce. For this reason, we adapt the uniformity principle to the multimodal setting by applying it not to individual embeddings, but to the centroids of aligned samples, that is, to the average representations of semantically matching pairs across modalities. This design retains the benefits of uniform coverage of the latent space while avoiding crucial interference with alignment. The L_CU loss encourages the centroids of aligned multimodal samples to be well-separated on the hypersphere, effectively ensuring that different semantic concepts remain distinguishable. At the same time, the L_ATP loss guarantees that all modalities representing the same semantic concept are aligned.
> > > To prove the limitations of the uniform spread of all embeddings, we perform an ablation study on the CIFAR10 dataset, in which we apply the uniformity loss proposed in (Wang & Isola, 2020) and our L_CU loss. As it is clear from the table below, without any other changes, the proposed L_CU improves the performance of both retrieval (R@1) and clustering (V-Measure) over the uniformity loss of (Wang & Isola, 2020).
> > >
> > > | Method                   | R@1       | V-Measure |
> > > |--------------------------|-----------|-----------|
> > > | L_CL+L_uniform (Wang & Isola) | 71.12     | 55.81     |
> > > | L_CL+L_ATP+L_uniform (Wang & Isola) | 82.47     | 64.72     |
> > > | L_CL+L_ATP+L_CU (ours)   | **84.64** | **71.47** |
> > >
> > > We report this ablation study in Appendix A.4 of the paper.
> > >
> > > 2. We would like to thank the Reviewer for raising this interesting question and for giving us the opportunity to clarify this aspect more thoroughly.
> > > In our framework, the contrastive losses involve only the classification tokens of the transformer models, thereby aligning the semantics of these tokens across the different modalities. Although the classification tokens influence all other tokens within a modality, they remain free to be learned in a way that supports downstream tasks.
> > > Furthermore, in our work, we evaluate downstream tasks that require redundant, unique, and synergistic information. Indeed, in tasks such as Multimodal (MM) Retrieval with cross-conditioning and Captioning in Tab. 2, the framework leverages all the hree types of information across all the tokens of all the modalities. By reducing the modality gap and better aligning the representations with our method, we observe clear performance improvements: +4.8 R@1 in Multimodal Retrieval compared to CLIP (LT) on MSCOCO, and +7.7 CIDEr in multimodal captioning on MSCOCO.
> > >
> > > We would like to thank the Reviewer once again for the insightful discussion that give us the opportunity to clarify some aspects of our paper.

---

### Author Response · Authors · 2025-11-28
**Gentle Reminder for Deadline Approaching**

Dear Reviewers,

this is just a gentle reminder that the rebuttal deadline is approaching.
We would like to thank the Reviewers for their effort, and we kindly ask to have a look at our responses as we have:

- Clarified the theoretical grounding of our method by including additional proofs on (i) Equation 6 on the gap affecting group-wise scattering, and (ii) the uniformity spread over the unit hyperphere of the RBF kernel of our loss.
- Performed additional experiments with novel comparisons (CoMM, 2025 and OTI/OVI, 2025), outperforming both approaches.
- Showed that the proposed method preserves modality-specific features by performing an additional experiment and explaining the reason why the proposed method is effective in doing so.
- Showed and explained the novelty and the effectiveness of the proposed centroid uniformity loss over previous approaches through a discussion and an ablation study.
- Performed suggested ablation studies on: (i) the gap at the initialization and (ii) on the ATP loss collapse.
- Plotted and clarified the rise of semantic stripes in the presence of gap.
- Theoretically clarified and discussed several aspects of our paper by providing rigorous proofs and thorough discussions of each point the Reviewers raised.


If any further concerns remain, we would be more than happy to discuss them with the Reviewers.

Thank you for your time and effort,

The Authors

---

### Meta-Review · Area_Chair_9eBw · 2026-01-07

**Summary:**

This paper proposes an approach to address the modality gap by introducing two complementary loss functions. Based on this, they design a loss function which encourages groups of positive samples to be compactly embedded, while uniformly dispersing negative samples on the hypersphere.  Experiments on several bi- and tri-modal datasets demonstrate improved clustering performance and maintained retrieval accuracy.

The paper received comments from four reviewers. The authors responded to the reviewers' comments and had thorough discussions with some of them. Based on the discussions, the authors have resolved some of the reviewers' concerns.

**Reviewer Concerns:**

During the rebuttal phase, the authors addressed the following concerns:
- The motivation of the work, particularly inconsistencies between the identified problems and the proposed solutions;
- The rigor of certain conceptual definitions;
- The novelty of the proposed methods;
- The depth of the theoretical exploration.

**Reviewer Scores:**

The authors engaged in extensive discussions with Reviewers 4NhU, d752, and pS85. Based on the substance of these exchanges, the authors have addressed the majority of the concerns raised by the reviewers.

---

### Decision · Program_Chairs · 2026-01-26

Accept (Poster)